# CXCR4 mediated recognition of HIV envelope spike and inhibition by CXCL12

Zhiying Zhang[1,6] ✉, Hongwei Zhang[1,6], Lyuqin Zheng[2,6], Shihua Chen[3,4], Shuo Du [3,5], Junyu Xiao [3,4,5] ✉ & Dinshaw J. Patel [1] ✉

CCR5 and CXCR4 both act as HIV co-receptors, though CXCR4 is less explored. CXCR4 binds the chemokine CXCL12 to regulate cellular processes and mediate HIV entry, a process that CXCL12 inhibits. Using cryo-EM, we investigate HIV-2 envelope (Env) spike recognition by CXCR4 and how CXCL12 inhibit this interaction. We discover that CXCR4 unexpected forms a tetramer, both alone and in complex. It binds CXCL12 with 4:8 and 8:8 stoichiometries, with the CXCL12 N-terminus inserting into the CXCR4 pocket. Structures of CXCR4-gp120[HIV-2] complex show one or two gp120 molecules per CXCR4 tetramer, with the V3 loop occupying the major sub-pocket of CXCR4 through deep embedment of its GFKF motif. The CXCL12 N-terminus chashes with gp120[HIV-2] V3 loops, explain its inhibitory effect. Docking analyses of other HIV antagonists further clarify their mechanisms. The CXCR4-gp120[HIV-1] model illustrate how V3 loop residues define co-receptor specificity, offering insights into co-receptor switching and therapeutic design.

Chemokine receptors are a family of G protein-coupled receptors (GPCRs) composed of seven transmembrane domains that mediate cell trafficking and adhesion during inflammation, routine immune surveillance, and development[1]. The C-X-C chemokine receptor type 4 (CXCR4) is a Gα$_i$-coupled GPCR and is activated exclusively through binding to its cognate ligand CXCL12 (also known as stromal cell-derived factor, SDF-1)[2,3]. Global deletion of CXCR4 in mice is embryonic lethal[4,5]. CXCR4 continues to play a crucial role in regulating vascular development, hematopoiesis, and the central nervous system[6,7]. The role of CXCR4 in cancer is much broader, given that it has been identified as a cancer marker to promote cancer cell migration and invasion[8–12].

In earlier studies, the chemokine receptor CXCR4 and CCR5 were discovered to function as co-receptors for the HIV infection[13,14], and the binding of endogenous ligand CXCL12 to CXCR4 was found to inhibit HIV entry[2,3]. HIV is characterized by two subtypes: HIV-1 and HIV-2. Entry of HIV into a host cell depends on a heavily glycosylated homotrimer consisting of the receptor-binding subunit gp120 and the fusion subunit gp41[15]. The entry process begins with the binding of the gp120 subunit to the primary receptor CD4, which triggers structural rearrangements in gp120 that exposes its V3 loop for co-receptor binding[16–19]. The subsequent conformational changes in the fusion subunit gp41 bring the viral and host membranes together, promoting their fusion[20].

As co-receptors for HIV, CXCR4 and CCR5 share several common features: an extracellular N-terminal domain, three extracellular loops (ECL1-3), and three intracellular loops (ICL1-3). The cryo-EM structure of the CCR5-gp120-CD4 complex reveals that the N-terminus of CCR5 wraps around gp120, while the V3 loop of gp120 inserts into the binding pocket of CCR5, which is formed by the bundle of seven transmembrane helices[21]. Site-directed mutagenesis studies have indicated that the N-terminal domain and ECL2 of CCR5 mediate the interaction with gp120, consistent with the cryo-EM structure of the complex[22]. By contrast, the ECL3 of CXCR4 has been suggested to further strengthen the stabilization of CXCR4-gp120 complex[23].

Early X-ray structures established that CXCR4 forms homodimers and forms complexes with a 2:1 protein: ligand stoichiometry when bound to the antagonist small molecule IT1t and the cyclic peptide

[1]Structural Biology Program, Memorial Sloan-Kettering Cancer Center, New York, NY, USA. [2]Molecular Biology Program, Memorial Sloan-Kettering Cancer Center, New York, NY, USA. [3]State Key Laboratory of Protein and Plant Gene Research, Peking University, Beijing, China. [4]Peking-Tsinghua Center for Life Sciences, Peking University, Beijing, China. [5]School of Life Sciences, Peking University, Beijing, China. [6]These authors contributed equally: Zhiying Zhang, Hongwei Zhang, Lyuqin Zheng. ✉e-mail: zhangz7@mskcc.org; junyuxiao@pku.edu.cn; pateld@mskcc.org

CVX15[24]. Recent cryo-EM structures establish that CXCR4 can adopt trimeric and tetrameric topologies in the presence of inhibitory antibodies[25], a finding that aligns with an earlier biochemical study[26]. Nevertheless, the structure of CXCR4 bound to HIV envelope (Env) spike remains unknown, likely due to the complicated oligomerization states of CXCR4, which might make it difficult to bind gp120, a challenge yet unresolved even three decades after its discovery as a HIV co-receptor[13,14]. By contrast, CCR5 co-receptor functions primarily as a monomer, and mutantions in specific residues can drive co-receptor switch[27–31].

In this work, we present a series of cryo-EM structures of CXCR4, including CXCR4 tetramer, in complex with its cognate ligand CXCL12, as well as in complex with the HIV-2 envelope spike gp120 alone, and in combination with CD4. Our data provide a structural basis for CXCR4 ligand recognition, insights into co-receptor switching, CXCL12-based HIV entry inhibition, HIV envelope spike recognition, and the rational design of therapeutic agents targeting the CXCR4 pocket so as to inhibit HIV infection.

## Results

### Overall structure of human CXCR4 tetramer

To obtain the human CXCR4 structure, we fused a Flag-tag at the C terminus of the full-length optimized coding DNA sequence. The purification process began with affinity chromatography, followed by gel filtration after expression in the FreeStyle™ 293-F cells. The elution volume of gel filtration suggested that CXCR4 primarily exists as an oligomer. Additionally, we observed a minor peak following the main peak. SDS-PAGE analysis confirmed that both peaks correspond to CXCR4 protein (Supplementary Fig. 1a–c). Native page showed that the minor peak has an apparent molecular weight of approximately 120 kDa, while the main peak is around 480 kDa−roughly four times larger. Based on these observations and consistent with a recent publication[25], we assigned these two peaks to monomeric and

tetrameric forms of CXCR4, respectively (Supplementary Fig. 1d, e). The rising molecular weights observed on the native gel are likely influenced by detergent micelles surrounding the CXCR4 proteins.

On the SDS page, we observed diffuse CXCR4 bands, which shifted to a low molecular weight upon treatment with PNGase F−an enzyme for removing almost all N-linked oligosaccharides from glycoproteins, suggesting that CXCR4 is highly glycosylated (Supplementary Fig. 1f). Preliminary cryo-EM analysis revealed that the CXCR4 particles exhibited severe preferred orientation on holey-carbon gold grids (Quantifoil Au 300 mesh R1.2/1.3). To address this issue, we added 0.5 mM detergent octyl maltoside fluorinated (FOM) to the solution prior to vitrification of cryo-grids. Through a cascade of two-dimensional (2D) and 3D classification procedures, a non-uniform refinement of 734,989 particles yielded a 2.9-Å density map (Supplementary Fig. 2a–c). This map enabled us to reconstruct nearly the full-length of the CXCR4 structure, except for one region: the N-terminus of CXCR4 (residues 1–24), which remains unresolved due to its intrinsic flexibility, likely related to its role in CXCL12 binding[32]. 2D classification identified a minor population (-3%) of monomeric CXCR4 (Supplementary Fig. 2d). However, the limited particle number and size constraints of cryo-EM prevented us from reconstructing a high-resolution structure.

The CXCR4 structure adopts a homotetramer topology with C4 symmetry (Fig. 1a). The tetrameric assembly is primarily mediated by interactions between transmembrane helices 5, 6, 7 (TM5, TM6, TM7) from one protomer and TM1 of an adjacent protomer (Fig. 1b). While hydrophobic interactions (Fig. 1c), typically considered the main driving force of transmembrane assembly, play an important role, a few polar residues near the end of helices form stable hydrogen bonds, further strengthening the organization of CXCR4 tetramer (Fig. 1c). On the extracellular side, Q272 from TM6 and E275 from TM7 establish interactions with K38 and N35 of TM1, respectively (Fig. 1c). On the intracellular side, two polar interactions are observed between TM1

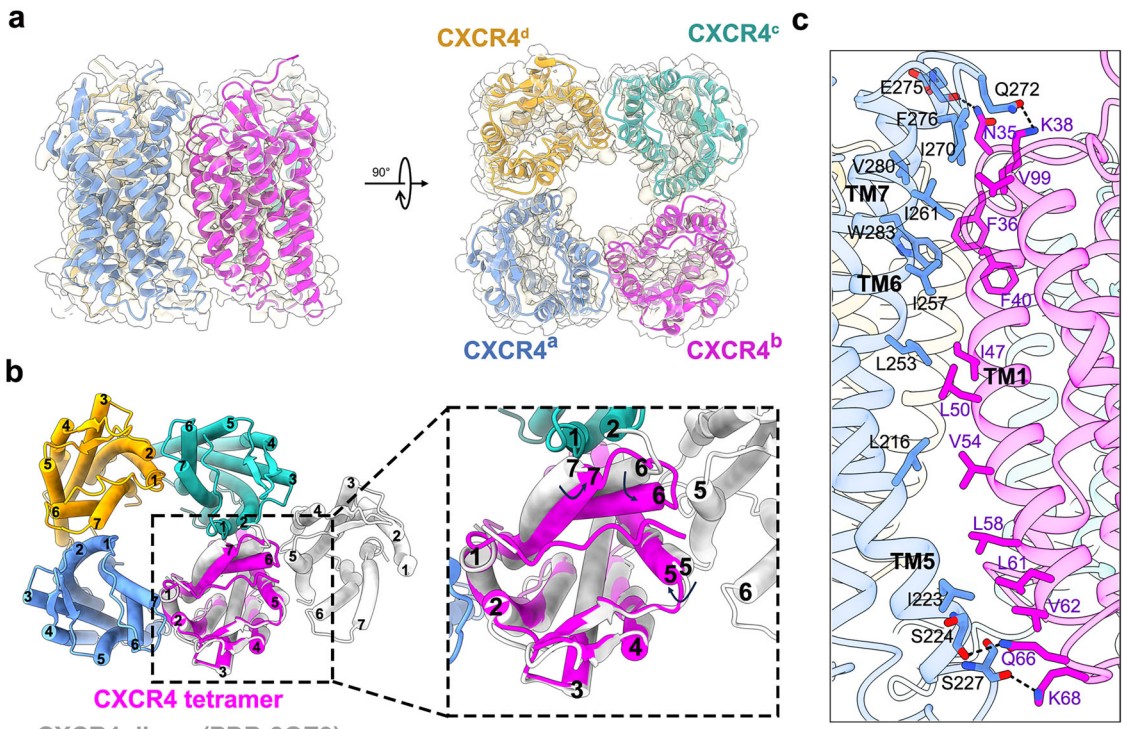

**Fig. 1 | Overall cryo-EM structure of CXCR4 tetramer. a** Different views of CXCR4 tetramer, with the four promoters are shown in blue, magenta, green, and yellow. **b** Comparison of the CXCR4 dimer interface in the CXCR4 tetramer (in color) and

the CXCR4 crystal structure (PDB:3OE9) (in silver). **c** Detailed interactions between CXCR4 monomers at the tetramer interface.

(Q66, K68) and TM5 (S224, S227) (Fig. 1c). Additionally, π−π interactions also contribute to the stability of the tetramer architecture, with F36 and F40 of TM1, along with W283 of TM7, forming key interfaces between CXCR4 monomers (Fig. 1c). The interface between two monomers in the cryo-EM-based tetrameric alignment differs notably from that in the crystal structure-based dimeric alignment of CXCR4 (Fig. 1b)[24], with the latter assembled through interactions between two TM5 helices. The structure of CXCR4 tetramer is nearly identical to previously reported tetrameric CXCR4 structures (PDB: 8YU7 and 8U4T)[25,33], with root mean square deviation (RMSD) values of 0.840 Å and 0.871 Å, respectively. Only minor differences are observed in the N-terminus and the ECL2 region of CXCR4, which are responsible for the interactions with the Fab (Supplementary Fig. 2e). Alignment of one CXCR4 monomer of the tetrameric and dimeric structures reveal a significant movement in TM6 and TM7, with a rotation of 10° in TM6 and 5° in TM7 (Fig. 1b). In summary, these structural differences highlight an alternative assembly mode for CXCR4 and provide a plausible explanation for oligomer switching, as CXCR4 has been shown to exist as a monomer when in complex with the downstream factor Gi proteins[25,34].

## Two distinct assembly mechanisms of CXCL12-CXCR4 complex formation

To understand how the cognate ligand CXCL12 binds to its receptor CXCR4 and the mechanism by which it blocks HIV entry, we coexpressed CXCL12 and CXCR4 that yielded a single peak on a size-exclusion column (Supplementary Fig. 3a) and determined the structure of the complex. We obtained two distinct structures of the CXCL12-CXCR4 complex (Supplementary Fig. 3b–d), namely with 8:4 stoichiometry at 3.4 Å (Fig. 2a) and 8:8 stoichiometry at 3.3 Å (Fig. 2b). Both states were confirmed by native PAGE analysis (Supplementary Fig. 3e). CXCL12 has been reported to exist in monomeric and dimeric states, each leading to distinct signaling events[35]. Its binding to CXCR4 mirrors interactions observed for other C-X-C chemokines and receptors in the presence of Gi proteins during activation[25,34,36,37]. In the CXCL12-CXCR4 structures reflecting the inactive state, CXCR4 adopts a tetrameric state (Fig. 2a), with each tetramer associated with four CXCL12 dimers (Fig. 2a). Interestingly, the four CXCL12 dimers can further attach to another CXCR4 tetramer, forming a "head-to-head" assembly configuration (Fig. 2b). Like the CXCR4 tetramer alone, the CXCL12-CXCR4 complexes also adopt C4 symmetry, with a RMSD of 0.658 Å in CXCR4 tetramer regions (Supplementary Fig. 4a).

The interaction of CXCL12 and CXCR4 is in line with a "two-site" model of chemokine binding. The N-terminus of CXCR4 acts as the chemokine recognition site 1 (CRS1), engaging with the globular core of CXCL12, while the seven transmembrane helices of CXCR4 form chemokine recognition site 2 (CRS2), which accommodates the insertion of the N-terminus of CXCL12 into a pocket[38–40]. In our structure, similar to the CXCR4 tetramer alone, the N-terminal residues (1–25) of CXCR4 in the CXCL12-CXCR4 complex are not visible, which is consistent with all reported CXCR4 ligand-bound and apo states, including complexes with chemokines (PDBs: 8U4O and 8K3Z)[25,34], viral peptides (e.g., vMIP-II, PDB: 4RWS)[41], small-molecule antagonists (PDBs: 8ZPL, 8ZPM, 8ZPN)[42]. At the CRS2 recognition site, K1 of CXCL12 forms a salt bridge with E288 of CXCR4, along with possible cation-π interactions involving $Y116_{CXCR4}$ and $W94_{CXCR4}$ (Fig. 2c, boxed segment). Additionally, $V3_{CXCL12}$ and $L5_{CXCL12}$ engage in hydrophobic interactions with the $I259_{CXCR4}$ and $I284_{CXCR4}$ (Fig. 2c, boxed segment). Within the ECL2 of CXCR4, two key interaction sites are observed between CXCL12 and the ECL2 of CXCR4, which is designated CRS3. The first involves $Y7_{CXCL12}$, which stacks against $F189_{CXCR4}$ and $Y190_{CXCR4}$ through π-π interactions (Fig. 2c, boxed segment). The second site involves the curved loop between β1 and β2 of CXCL12, where the main chain probably contacts with $D181_{CXCR4}$ (Fig. 2c, boxed segment). On the opposite side, in the ECL3 region, $E268_{CXCR4}$ forms a

single salt bridge with $R12_{CXCL12}$, further supported by interactions with the main chains of $I265_{CXCR4}$ and $L266_{CXCR4}$ (Fig. 2c, boxed segment), consistent with the mutagenesis studies highlighting the importance of this region for CXCL12 binding[23,43]. In terms of the CSR1, $E26_{CXCR4}$ is sandwiched between the N-loop and β3 regions of CXCL12, reinforcing the binding between CXCL12 and its receptor (Fig. 2c). Mutagenesis experiments reveal that single mutation at K1, S6 or Y7 in CXCL12 have minimal effects on CXCL12-CXCR4 binding (Fig. 2d). In contrast, a mutation at R12, or simultaneous mutations in all interaction residues, completely abolish interactions between CXCL12 and CXCR4 (Fig. 2d). The $R12_{CXCL12}$ mutation was also reported to cause a dramatic loss of potency in CXCR4-mediated intracellular calcium signaling and chemotactic responses[44-46]. These findings highlight the critical role of the ECL3 region of CXCR4 in mediating its interaction with CXCL12.

Comparison of CXCL12-CXCR4 complexes in the inactive and active (PDB: 8K3Z) states reveals two striking divergences. One, as expected from GPCR activation[47], TM5-7 undergo notable shifts (approximately 12 Å in TM5, 13 Å in TM6 and 10 Å in TM7) between these two states (Fig. 2e). The other, the relative orientation of CXCL12 and CXCR4 is rotated, with a displacement of 7 Å in the active state compared to the inactive state (Fig. 2e). This shift alters the conformation of CXCL12, particularly the N-terminal loop (Fig. 2e), which corresponds to different binding interfaces between these two states[34]. The CXCL12 dimer interface in our structures is primarily mediated by main-chain interactions between the β1 strands of the two monomers (Supplementary Fig. 4b, c), closely resembling both the crystal and NMR structures[48–50] (PDB IDs: 1QG7, 1A15, and 2K01, corresponding to a mammalian crystal form, a synthetic variant, and an NMR model, respectively). The CXCL12 dimer in our structure closely resembles, except for two regions: the N-terminal segment, which is critical for receptor interaction, and the C-terminal helix (Supplementary Fig. 4d).

## Architecture of CXCR4-gp120[HIV-2] complex

The first step in reconstituting the CXCR4-gp120 complex involved purifying the CXCR4 and gp120 independently (see "Methods"). We selected gp120 from HIV-2 instead of HIV-1 due to its higher binding affinity for CXCR4[23]. CXCR4 and gp120 were mixed at a 1:2 molar ratio before being subjected to size-exclusion chromatography (SEC). The peak corresponding to CXCR4-gp120[HIV-2] complex (Supplementary Fig. 5a) was collected and concentrated for preparation of cryo-grids. Structural data processing revealed CXCR4-gp120[HIV-2] complex exhibits multiple conformations (Fig. 3a and Supplementary Fig. 6), with the relative position between CXCR4 and gp120[HIV-2] being dynamic. This flexibility results in various alignments of the CXCR4-gp120[HIV-2] complex. Local refinement of the gp120[HIV-2] region enabled the reconstruction of the entire V3 loop and docking of most of the gp120[HIV-2] structure (Fig. 3b, c). Notably, the V3 loop shows minimal changes, amongst conformers, indicating a conserved mechanism of gp120[HIV-2] binding to its co-receptor CXCR4. The CXCR4 and gp120[HIV-2] complex exhibits a 4:1 stoichiometry (Fig. 3a, b). A small number of particles containing two gp120[HIV-2] proteins were also observed in the 2D classifications (Supplementary Fig. 6b, boxed segment), but these particles could not be resolved to high resolution. Structural modeling suggests that the CXCR4 tetramer can accommodate four gp120[HIV-2] proteins without any apparent steric clashes (Supplementary Fig. 7a), indicating that the complex with a single gp120[HIV-2] bound may be more stable than other configurations, at least when only gp120[HIV-2] is present.

For these structures, residues 305–313 of gp120[HIV-2] are buried into the CXCR4 CRS2 pocket, adopting a reverse cross conformation (Fig. 3d). In the turn region of the V3 loop, residues F309, K310, and F311 form the GFKF motif, alongside G308, which is essential for HIV-2 infection (Fig. 3d). F309 is the most deeply embedded, engaging in

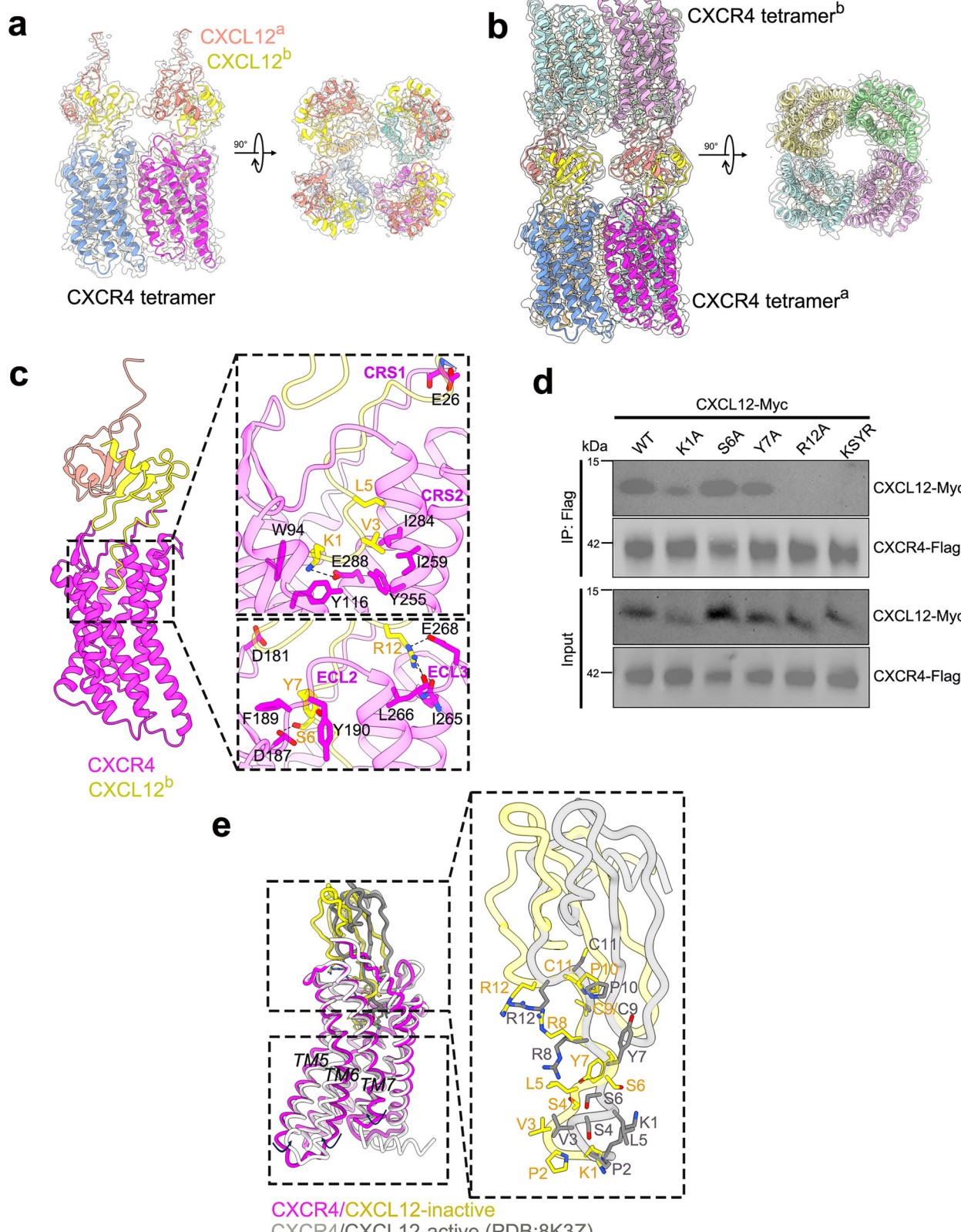

**Fig. 2 | Overall cryo-EM structure of CXCL12-CXCR4 complex. a** Different views of CXCL12-CXCR4 complex with 8:4 stoichiometry. **b** Different views of CXCL12-CXCR4 complex with 8:8 stoichiometry, with the two head-to-head CXCR4 tetramers labeled as tetramer[a] and tetramer[b]. **c** Detailed interactions between CXCL12 and CXCR4 with residues labeled in the boxed panels. **d** The CXCL12 mutants exhibited either similar or impaired binding to CXCR4. The Co-immunoprecipitation (Co-IP) experiments were repeated at least two times with similar results. Source data are provided as a Source data file. **e** Structural comparison of CXCL12-CXCR4 complex between inactive (in color) and active (in silver) (PDB:8K3Z) states.

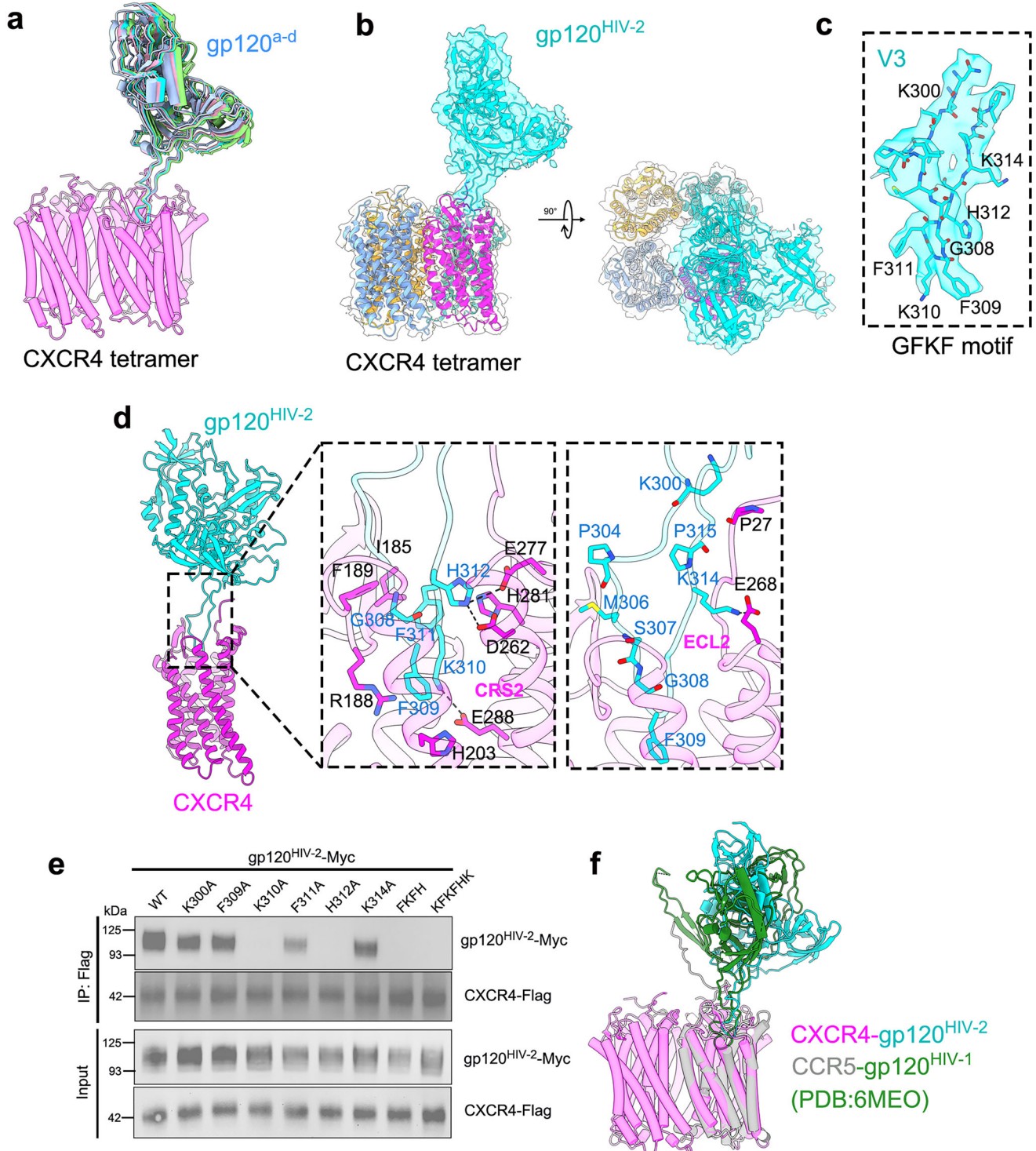

**Fig. 3 | Overall cryo-EM structure of CXCR4-gp120^HIV-2 complex. a** Structural diversity of the CXCR4-gp120^HIV-2 complex, superscripts^a–d indicate distinct conformations with slight structural variations. **b** Different views of CXCR4-gp120^HIV-2 complex, highlighting the V3 loop of gp120. **c** Highlighted key residues within the V3 loop that are critical for CXCR4 recognition, including the labeled GFKF motif. **d** Detailed interactions between HIV-2 gp120 and CXCR4. Interfacial residues are labeled in the boxed segment. **e** The gp120^HIV-2 mutants exhibited either similar or impaired binding to CXCR4. The pull-down assays were repeated at least two times with similar results, and source data are provided as a Source data file. **f** Structural comparison of CXCR4-gp120^HIV-2 (in pink and cyan) and CCR5-gp120^HIV-1 (PDB:6MEO) (in silver and green) complexes.

both a cation-π interaction with R188_CXCR4 and a π-π interaction with H203_CXCR4. K310 forms a salt bridge with E288_CXCR4, while F311 establishes local hydrophobic interactions with I185 and F189 in the ECL2 region of CXCR4 (Fig. 3d). At CRS2, H312 plays a critical role by making extensive contacts with neighboring CXCR4 residues. It forms

a bifurcated hydrogen bond with both E277_CXCR4 and D262_CXCR4, and a π−π interaction with H281_CXCR4 (Fig. 3d, left boxed segment). In particular, the loop formed by residues M306-F309 of gp120 comes into close proximity with the ECL2 region of CXCR4, producing main-chain contacts. In the ECL3 region, K314 interacts with E268_CXCR4 through a

salt bridge (Fig. 3d, right boxed segment), which further stabilizes the V3 loop. K300, situated above the inserted loop, is positioned near the N-terminus of CXCR4, suggesting potential interactions due to their proximity (Fig. 3d, boxed segment). However, mutating lysine to alanine does not affect the gp120[HIV-2]-CXCR4 binding (Fig. 3e). Further mutational analyses highlight the functional significance of key residues. Single mutations at K310 or H312 disrupt gp120[HIV-2]-CXCR4 interactions, while mutations at F309 and F311 do not, indicating the critical roles of K310 and H312 in assembling the gp120[HIV-2]-CXCR4 complex, which closely corresponds to the key hydrogen bonds formed between K310, H312, and CXCR4. Combined mutations of residues F309-H312 or an expanded set including K300, F309-H312, and K314 effectively block the interaction between gp120 and CXCR4 (Fig. 3e). Our structural model also reveals that the N-terminus of CXCR4 runs parallel and anti-parallel to the entry and exit loops of the V3 region of gp120[HIV-2] (Fig. 3d). These three loops are tightly packed, making significant connections with one another (Fig. 3d). Furthermore, we noticed three proline residues-P304, P315 in gp120, and P27 in CXCR4-closely positioned within these loops, potentially altering the loop direction (Fig. 3d, boxed segments). This structural arrangement could explain the dynamics of the gp120[HIV-2] binding, consistent with the multiple observed structural conformations. The CXCR4 residues involved in binding HIV-2 gp120 are similar to those involved in HIV-1 gp120 binding[23,51], suggesting a conserved mechanism for both HIV-1 and HIV-2 infection.

To determine whether the CXCR4 pocket in this configuration could accommodate the V3 loop of HIV-1, we compared our structure to the CCR5-gp120[HIV-1] complex by aligning the CXCR4 and CCR5 protomers (Fig. 3f). As both CXCR4 and CCR5 serve as co-receptors for HIV, they adopt a similar configuration, with an RMSD of 1.131 Å. This close alignment suggests that the V3 loop of HIV-1 is also well-suited for binding to the CXCR4 pocket. Structural and sequence alignments reveal that the key residues involved in HIV-1 binding to CCR5 are largely conserved in CXCR4 (Fig. 3f and Supplementary Fig. 8a). This conservation suggests that the combination of HIV-1 gp120 and CXCR4 may closely resemble the structure of the CXCR4-gp120[HIV-1] complex. The binding pocket of chemokine receptors is typically divided into two sub-pockets: the minor sub-pocket, formed by TM1-3 and TM7, and the major sub-pocket, formed by TM3-7[52]. Interestingly, the V3 loop of HIV-1 primarily occupies the minor sub-pocket, while the V3 loop of HIV-2 predominantly fills the major sub-pocket (Fig. 3d). This distinction may arise from the absence of two residues in the HIV-2 V3 loop or from the sequence diversity in this region (Supplementary Fig. 8b). Another notable feature is the relative position of gp120 in HIV-1 compared to HIV-2. In HIV-1, gp120 exhibits a significant shift of 12 Å and a clockwise rotation of 150°, making its position closer to the center of the CXCR4 tetramer, particularly around the bridging sheet, where it contacts the N-terminus of CCR5. This movement also results in steric clashes in the bridging sheet region, making it unlikely that four, or even two, HIV-1 gp120 proteins could bind the CXCR4 tetramer simultaneously (Supplementary Fig. 7b). However, we cannot entirely rule out the possibility of four HIV-1 gp120 proteins binding, as structural rearrangements in HIV-1 gp120 could potentially accommodate such recognition (Supplementary Fig. 7b).

### Reconstitution of CXCR4-gp120[HIV-2]-CD4 complex
The mature viral spike contains three copies of gp120 (gp120$_3$), which potentially engage three copies of CD4. However, the extent of CD4 binding leads to distinct conformational outcomes. When no or only one CD4 molecule is bound, the trimeric Envs remain in a closed, prefusion state. By contrast, binding of two or three CD4 molecules induces an open state in the Env trimer[16,53,54]. This asymmetrical engagement of CD4 molecules has also been observed in native membranes[55]. In the CCR5-gp120[HIV-1]-CD4 structure, no significant differences were observed in the core region of gp120 in the presence of

CCR5[21]. It is essential to determine whether this holds true for the CXCR4-gp120[HIV-2]-CD4 structure, as HIV-1 and HIV-2 exhibit notable differences in gp120 configuration when bound to CXCR4 (Fig. 3f). To obtain the CXCR4-gp120[HIV-2]-CD4 complex, we first expressed the extracellular D1-D4 domains of CD4 and incubated them with gp120. The mixture was then combined with CXCR4 to form the CXCR4-gp120-CD4 complex (Supplementary Fig. 5b). The resulting structure, at a modest 5.6 Å resolution, enabled us to dock the CXCR4-gp120 complex and the D1-D2 domain of CD4 (Fig. 4a, Supplementary Fig. 9). Unlike the CXCR4-gp120 complex, most of the particles of CXCR4-gp120-CD4 complex displayed two gp120 molecules bound, as indicated by the 2D classification (Supplementary Fig. 9b). However, the density map revealed that only one half of the gp120-CD4 complex was well-resolved, while the other half showed only part of the gp120 molecule. This contrasts with the CXCR4-gp120 complex alone, where most particles had only one bound gp120. Native page gel results showed a broad distribution of CXCR4-gp120[HIV-2] and CXCR4-gp120[HIV-2]-CD4 complexes (Supplementary Fig. 5c, d), making it difficult to determine the exact stoichiometry of gp120 and CD4 binding to CXCR4 tetramer. To further clarify this, we employed mass photometry to assess their molecular weights (Supplementary Fig. 5e–g). Despite the heterogeneity, the main peak suggested that one or two gp120 molecules, along with one or two CD4 molecules, bind to CXCR4 tetramer, consistent with cryo-EM observations. One possible explanation for this structural variability is that CD4 binding may increase the stability of gp120, favoring the presence of two gp120 molecules in the complex.

Alignment of CXCR4-gp120 and CXCR4-gp120-CD4 complexes shows no obvious conformational changes in gp120 upon CD4 binding (Fig. 4b). Comparison of our structure to CD4-bound SOSIP Env trimer reveals no significant differences in the core region of gp120, consistent with the CCR5-gp120[HIV-1]-CD4 structure. The HIV-1 Env trimer could align with three gp120 molecules from the CXCR4$_{tetramer}$-gp120[HIV-2]-CD4 complexes without experiencing any spatial clashes (Fig. 4c). To investigate whether this compatibility extends to HIV-1 gp120, we modeled the Env trimer with three generated CXCR4-gp120[HIV-1]-CD4 complexes. Surprisingly, the model revealed dominant clashes within adjacent CXCR4 tetramers, preventing the trimeric Env from simultaneously binding three CXCR4-gp120[HIV-1]-CD4 complexes (Fig. 4d). This structural incompatibility highlights the notable differences between HIV-1 and HIV-2 gp120 when bound to CXCR4, suggesting that HIV-2 may adopt a distinct assembly architecture during host cell infection. This difference may contribute to the divergent infection mechanisms of HIV-1 and HIV-2, reflecting the unique ways these viruses engage their co-receptors for efficient entry.

### Potential mechanism of co-receptor switch
It is well established that the amino acid sequence of V3 determines HIV-1 co-receptor usage-whether the virus binds to CCR5 ("R5 viruses"), predominantly infecting macrophages, or to CXCR4 ("X4 viruses"), primarily infecting T cells[29–31]. We revisited the modeled CXCR4-gp120[HIV-1] complex to investigate how a specific mutation could drive the transition from an R5 to an X4 phenotype. Previous studies have reported that mutations such as D320R or S306R in the V3 loop are associated with enhanced CXCR4 binding. In our structural model, this aspartate-to-arginine substitution enables a salt bridge with E31, located in the N-terminus of CXCR4, which strengthens the gp120-CXCR4 interaction (Fig. 5a, b). Similarly, the S306R introduces a positively charged residue that can form a cation-π with F189$_{CXCR4}$, further stabilizing the interaction (Fig. 5c). Residues 323 and 324 in the V3 loop, which are typically negatively charged or uncharged and associated with CCR5 usage, may from hydrogen bonds or salt bridges with D181 in the ECL2 region of CXCR4 when mutated to positively charged residues (Fig. 5d). Notably, these residues are not conserved in CCR5, highlighting their roles in co-receptor specificity and V3 loop

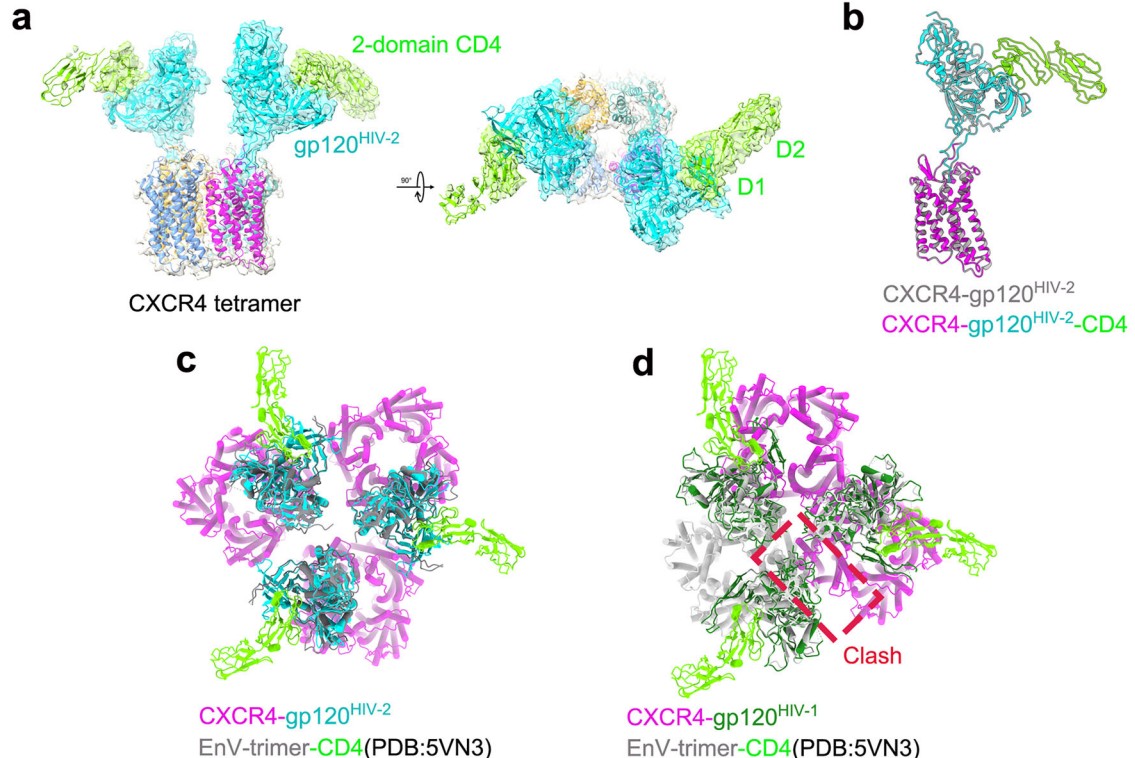

**Fig. 4 | Overall cryo-EM structure of CXCR4-gp120^HIV-2-CD4 complex. a** Different views of CXCR4-gp120^HIV-2-CD4 complex. **b** Structural comparison of CXCR4-gp120^HIV-2 (in silver) and CXCR4-gp120^HIV-2-CD4 (in color) complexes. **c** Structural modeling of high-order CXCR4-gp120₃^HIV-2-CD4₃ complex. **d** Structural modeling of high-order CXCR4-gp120₃^HIV-1-CD4₃ complex. Steric clashes are circled with a red dashed rectangle.

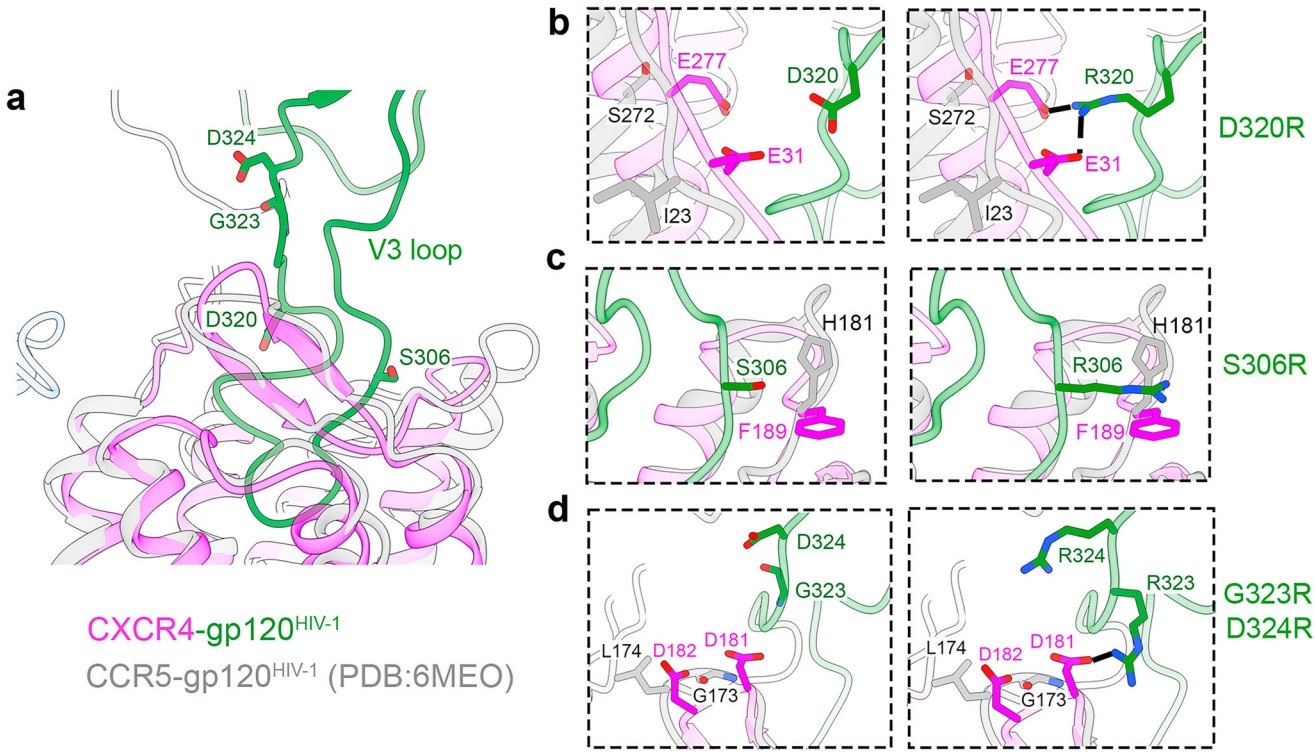

**Fig. 5 | Structural basis for enhanced CXCR4 binding by HIV-1 gp120 V3 loop mutants. a** Structural model of HIV-1 gp120 (green) bound to CXCR4 (magenta), overlaid with CCR5-bound gp120^HIV-1 from PDB: 6MEO (gray), highlighting the V3 loop and positions of key interface residues. **b**–**d** Close-up comparisons of wild-type (left panels) and mutant (right panels) V3 loop residues and their interactions with CXCR4. **b** The D320R mutation introduces a bifurcated salt bridge with CXCR4 residues E31 (N-terminus) and E277 (ECL3), enhancing electrostatic complementarity. **c** The S306R mutation enables a cation-π interaction with CXCR4 F189. **d** Double mutation G323R/D324R allows formation of hydrogen bonds or salt bridges with CXCR4 D181 and D182 in ECL2.

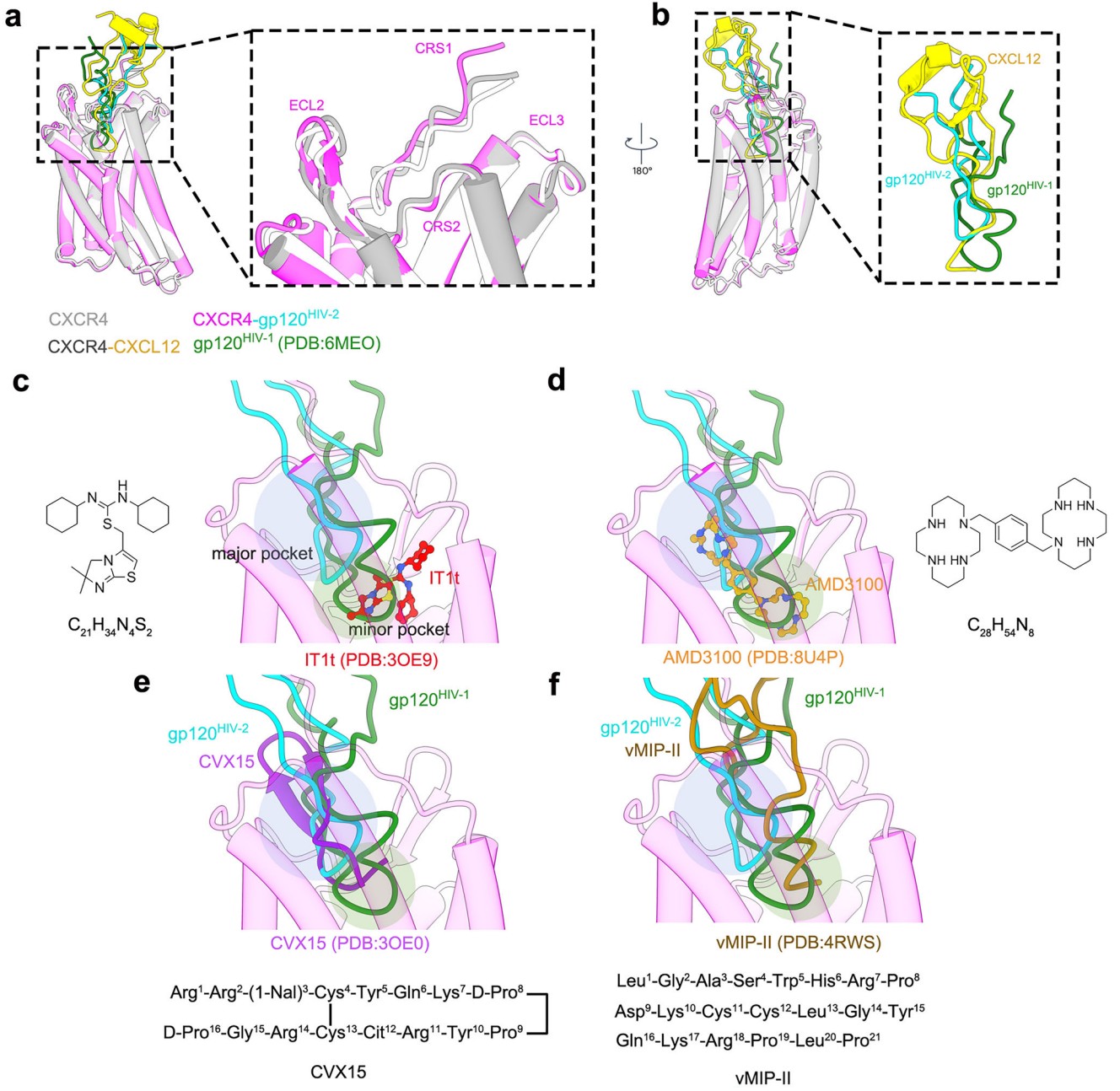

**Fig. 6 | CXCL12 and HIV antagonists. a** Comparison between the CXCR4 tetramer, the CXCL12-CXCR4 complex and the CXCR4-gp120^HIV-2 complex. Segments are labeled in the boxed panels. **b** Comparison between CXCL12 and the V3 loop of HIV-1 and HIV-2. Segments are labeled in the boxed panels. Structural modeling of the V3 loops of HIV-1, HIV-2, and other known CXCR4 and small molecule bound structures: **c** IT1t (PDB:3OE9); **d** AMD3100 (PDB:8U4P); **e** CVX15 (PDB:3OE0), and **f** vMIP-II (PDB:4RWS), the minor and major pockets of CXCR4 are highlighted with green and blue circular shading, respectively.

recognition. Our structural model offers a mechanistic explanation for how these site-specific mutations influence co-receptor preference, consistent with prior experimental observations[27,28,30]. It also provides fundamental structural evidence for how a single mutation can enable the switch from CCR5 to CXCR4 usage, contributing to our understanding of viral tropism and pathogenesis.

**Insight into CXCL12 and HIV antagonists' inhibition**

Our structures reveal that CXCR4, whether alone, in complex with CXCL12, or bound to HIV-2 gp120, undergoes minimal rearrangement at its four key binding sites: CRS1, CRS2, ECL2, and ECL3. Notably, the N-terminus of CXCR4 exhibits an outward movement when comparing the CXCL12-bound or gp120-bound states to CXCR4 alone (Fig. 6a).

This shift likely accommodates the ligand binding or HIV recognition, providing additional space for their loop insertion. Interestingly, the inserted loops of CXCL12 and HIV-1 penetrate roughly twice as deep as that of HIV-2 (Fig. 6b). Furthermore, the N-terminal loop of CXCL12 passes through the major sub-pocket to reach the minor sub-pocket, with its terminal residue K1 extending into the minor sub-pocket to form contacts with surrounding residues. The N-terminus of CXCL12 also exhibits extensive spatial clashes with the V3 loop of both HIV-1 and HIV-2 (Fig. 6b), explaining why CXCL12 is capable of inhibiting HIV infection by disrupting its binding.

To explore whether HIV antagonists inhibit both HIV-1 and HIV-2 entry, we analyzed the relative spatial positioning of the V3 loops of HIV-1 and HIV-2 in the presence of various HIV antagonists, using

available structural data[24,25,41]. For the small molecule IT1t, it is located in the minor sub-pocket, which causes a clear clash with the V3 loop of HIV-1 but not HIV-2 (Fig. 6c), indicating strong inhibition of HIV-1 infection with a weaker effect on HIV-2. By contrast, the small molecule AMD3100 possesses two "wings" that occupy both the major and minor sub-pockets (Fig. 6d), suggesting its ability to inhibit both HIV-1 and HIV-2 infections. Additionally, a cyclic peptide CVX15 blocks the entry of HIV-1 gp120 by stacking primarily into the major sub-pocket, with its N- and C-termini stretching into the minor sub-pocket, effectively blocking the insertion of the V3 loops of both HIV-1 and HIV-2 (Fig. 6e). Similar to the cognate ligand CXCL12, the viral chemokine vMIP-II also obstructs HIV-1 and HIV-2 entry by creating substantial clashes with the V3 loop (Fig. 6f). This comparative analysis offers valuable insights into how the binding of these antagonists can interfere with viral entry by targeting conserved or distinct regions within the V3 loops of both HIV-1 and HIV-2, providing a foundation for the rational design of therapeutic agents that enhance inhibition efficiency against both viral strains.

## Discussion

As a co-receptor of HIV, CXCR4 has been extensively studied and is considered crucial for drug design. However, its function remains poorly understood due to a lack of structural data on the HIV CXCR4-gp120 complex. In this study, we report a series of CXCR4 structures, all of which adopt unanticipated tetrameric folds. This contrasts with previously reported crystal structures, where CXCR4 adopts a dimeric fold[24]. Notably, the monomer-monomer interfaces in our cryo-EM structures and those in the crystal structures are strikingly different (Fig. 1b). Recent studies have also shown that CXCR4 can exist in monomeric and trimeric forms[25,34], highlighting the highly dynamic nature of CXCR4 assembly. This structural flexibility may be linked to its functional diversity, as different CXCR4 conformations could correspond to distinct cellular outcomes.

In the CXCL12-CXCR4 structures, two distinct conformations of the complexes are observed. Interestingly, four CXCL12 dimers could bind to two CXCR4 tetramers, suggesting that dimeric CXCL12 may activate two cells simultaneously. This observation is consistent with earlier findings that CXCL12 can function in both monomeric and dimeric forms, leading to distinct signaling events[35]. Upon activation, CXCR4 transitions to a monomeric state. The presence of hetero-trimeric Gi proteins induces structural rearrangements in TM5-7 of CXCR4 and in the N-terminus of CXCL12. These changes alter the binding interface between CXCL12 and CXCR4, facilitating the receptor's transition to its active state and disrupting the CXCR4 tetramer into monomers. Structural analysis suggests that the tetrameric form of CXCR4 cannot accommodate four heterotrimeric Gi proteins, indicating a regulatory mechanism that converts tetrameric CXCR4 into its monomeric form upon activation.

We attempted to assemble the CXCR4-gp120[HIV-1]-CD4 complex, and the results from gel filtration reveal a clear shift upon combining the CXCR4 and CD4-gp120[HIV-1] complex, suggesting a successful assembly of the CXCR4-gp120[HIV-1]-CD4 complex (Supplementary Fig. 10a–c). This finding was further supported by SDS-PAGE, which revealed distinct bands corresponding to CXCR4, HIV-1 gp120, and CD4 (Supplementary Fig. 10d). Despite successful assembly, we were unable to obtain a high-resolution structure, likely due to the inherent dynamics of gp120 binding to CXCR4, and the lower binding affinity of CXCR4 for HIV-1 gp120 compared to HIV-2 gp120[23]. These dynamics are also apparent in our CXCR4-gp120[HIV-2] structures, where the relative positioning between CXCR4 and HIV-2 gp120 is flexible. This structural diversity of CXCR4-gp120[HIV-2] complex may stem from three proline residues that act as a "gate" for the V3 loop insertion. This flexibility may also facilitate the binding of additional gp120 molecules. By contrast, HIV-1 gp120 is proposed to bind only one molecule, as the large rotation in the core region of HIV-1 gp120 causes clashes in the

bridging sheet region when modeling the second gp120[HIV-1] within the CXCR4 tetramer. Additionally, the N-terminus of gp120[HIV-2], which is responsible for binding gp41 in the prefusion or CD4-bound Env[16,20,56–58], is not visible in our structure. Comparison between the Env-trimer-CD4 and our structures shows no significant changes in the core region of CD4, consistent with observations from the CCR5-gp120[HIV-1]-CD4[21] structure. The Env trimer is capable of binding three tetrameric CXCR4 complexes without any clear steric clashes, due to the presence of three copies of the gp120-CD4 complex. Furthermore, the CXCR4 tetramer can bind two gp120[HIV-2]-CD4 complexes, potentially recruiting an additional Env trimer at a diagonal position to form a high-order CXCR4-gp120₃[HIV-2]-CD4₃ complex (Supplementary Fig. 11a). This modeling was performed using only D1 and D2 domains of CD4, as clashes emerge when the full D1-D4 domains of CD4 are included (Supplementary Fig. 11b). However, we cannot completely rule out the possibility of forming a high-order CXCR4-gp120[HIV-2]-CD4₃ complex, particularly if conformational changes occur in the core region of gp120 relative to its V3 loop. Based on the CCR5-gp120[HIV-1]-CD4 structure, we could model the CXCR4-gp120[HIV-1]-CD4 complex because the interacting residues involved in CCR5 binding are largely conserved in CXCR4. In both our modeling and the published CCR5-gp120[HIV-1]-CD4 structure, the interaction sites with HIV-1 gp120 in CXCR4 or CCR5 contain CSR1, CSR2, and ECL2. In contrast, the interaction sites with HIV-2 gp120 in our structure include the CSR1, CSR2, ECL2, and ECL3. The absence of ECL3 interactions may partly explain the lower binding affinity of HIV-1 gp120 to CXCR4.

Our structures offer valuable insights for the development of vaccines and therapeutic agents targeting both HIV-1 and HIV-2 infections. Such small-molecule antagonists, involving configurations like AMD3100, which features two cyclam moieties located in the minor and major sub-pockets, respectively (Supplementary Fig. 12), are ideal, demonstrating their potential to inhibit both HIV-1 and HIV-2 entry. The blocking activity of cognate ligand CXCL12 and viral chemokine vMIP-II against HIV entry also provides useful clues for the design of small peptide antagonists (Supplementary Fig. 12). The N-terminus of these chemokines inserts deeply into the CXCR4 binding pocket. Small peptides mimicking these chemokines could be designed to bind in a similar manner, potentially serving as effective blockers of HIV-1 and HIV-2 infection by targeting the CXCR4 binding pocket.

## Methods

### Cell culture

FreeStyle 293-F cells (Invitrogen) and Expi293 cells (Invitrogen) were cultured in FreeStyle 293 Expression Medium (Gibco) or Expi293 Expression Medium (Gibco), respectively, at 37 °C in a shaking incubator with 8% $CO_2$ and 70% humidity.

### Protein expression and purification

For recombinant expression in 293F cells, the gene of full-length human CXCR4 was cloned into the pcDNA3.4 vector with a C-terminal Flag-tag. Genes of human CXCL12, CD4-extracellular domain (ECD, residues 26–388), gp120[HIV-1] (from the 92BR020 stain, residue 30–505, Uniprot: B2LT42)[21] and gp120[HIV-2] (from the VCP strain, residues 23–501, Uniprot: Q70145)[50,59] were fused with interleukin-2 (IL-2) secretion signal peptides at their N-terminus and His tags at their C terminus, and then cloned into pcDNA3.4 vector separately. The primers used in this study are provided in the Supplementary Data file. All genes were codon optimized and synthesized by IDT. All constructions were generated using Gibson assembly (NEB) and confirmed by sequencing. When the cell density reached $1.5 \times 10^6$ cells ml⁻¹, the CXCR4 plasmid was mixed with 40-kDa linear polyethylenimine (PEI, Polysciences) at a ratio of 1:2 and incubated for 30 min before transfection. Sodium butyrate (10 mM final) was supplemented into the culture 16 h after infection. Cells were cultured for another 2 days

before harvest. To produce the CXCR4-CXCL12 complex, two plasmids were coinfected at a ratio of 1:2. Cells were harvested 3 days after sodium butyrate addition. For secreted expression, cells at $1.0 \times 10^6$ cells ml$^{-1}$ were infected with CD4-ECD or gp120$^{HIV-2}$ (mixed with PEI at a 1:2.5 ratio) and cultured for additional 4 days.

Cells expressing CXCR4 were collected and resuspended in lysis buffer (25 mM HEPES, pH 7.5, 150 mM NaCl, cOmplete (EDTA-free) protease inhibitor). Cells were lysed by sonication, and then 1% (w/v) Lauryl Maltose Neopentyl Glycol (LMNG, Anatrace) and 0.1% (w/v) cholesteryl hemisuccinate (CHS, Anatrace) were added to solubilize proteins for 3 h at 4 °C. Subsequently, insoluble debris was cleared by centrifugation at $22,000 \times g$ for 1 h, while the supernatant was mixed with anti-Flag resin (Sigma) for 3 h. The resin was washed by the wash buffer I (25 mM HEPES pH 7.5, 150 mM NaCl, 0.5% LMNG and 0.05% CHS) followed by the wash buffer II (25 mM HEPES pH 7.5, 150 mM NaCl, 0.001% LMNG, 0.0001% CHS and 0.00033% Glyco-diosgenin (GDN, Anatrace)), and eluted by the wash buffer II supplemented with 0.2 mg ml$^{-1}$ 3×Flag peptide. The elution proteins were concentrated and further purified by SEC using a Superdex 200 Increase 10/300 GL column (GE Healthcare) pre-equilibrated in the wash buffer II. Fractions corresponding to the tetrameric peak were pooled and concentrated for cryo-EM sample preparation. The purification procedure of the CXCL12-CXCR4 complex was the same as CXCR4 alone, except that elution complexes were incubated overnight with additional 0.0025 mg ml$^{-1}$ commercial CXCL12 (R&D Systems) before being loaded onto a Superose 6 Increase 10/300 GL column (GE Healthcare) for SEC.

Secreted CD4-ECD and gp120$^{HIV-2}$ were purified from the expression media of cell culture. The supernatant was collected from culture and dialyzed against 100× excess of dialysis buffer (25 mM Tris, pH 8.0, 150 mM NaCl) for 48 h at 4 °C. After that, the supernatant was loaded onto a gravity column of Ni-NTA resin. The resin was washed with 25 mM Tris, pH 8.0, 150 mM NaCl, and 20 mM imidazole. Finally, the protein was eluted by wash buffer supplemented with 300 mM imidazole. Eluted proteins were concentrated and loaded onto a Superdex 200 Increase 10/300 GL column (GE Healthcare) pre-equilibrated with the wash buffer II. Peak fractions were collected and concentrated before being flash frozen in liquid nitrogen and stored at −80 °C.

To produce CXCR4-gp120$^{HIV-2}$-CD4 complex, CD4 and gp120$^{HIV-2}$ were first assembled on ice overnight at a 1.3:1 molar ratio. gp120-CD4 complex was purified by SEC using Superdex 200 Increase 10/300 GL column (GE Healthcare). Subsequently, CXCR4 was mixed with gp120$^{HIV-2}$-CD4 complex at a molar ratio of 1:1.3 for 2 h. Complexes were loaded onto a Superose 6 Increase 10/300 GL column. The fractions containing purified CXCR4-gp120-CD4 complexes were collected and concentrated for cryo-EM analysis. For the CXCR4-gp120$^{HIV-2}$ complex, CXCR4 was mixed with gp120$^{HIV-2}$ directly at a molar ratio of 1:2 on ice for 4 h. The samples were then fractionated on a Superose 6 Increase 10/300 GL column (GE Healthcare). The purified samples were pooled and concentrated prior to Cryo-EM grid preparation.

## Cryo-EM sample preparation and data collection
For the CXCR4 apo sample, purified human CXCR4 protein was concentrated to 10 mg ml$^{-1}$, and octyl maltoside fluorinated (FOM) was added freshly at a final concentration of 0.5 mM to mitigate preferred orientation issues. A 4 μl aliquot of the mixture was applied to glow-discharged holey-carbon gold grids (Quantifoil Au 300 mesh R1.2/1.3). The grids were blotted for 2 s at 100% humidity and 6 °C, then rapidly frozen in liquid ethane using the Vitrobot Mark IV (FEI). Grid screening and data collection were conducted using a 300-kV Titan Krios G3 electron microscope with a K3 direct detector, operated with EPU software (Thermo Scientific). Images were captured at a pixel size of 1.07 Å, with defocus values ranging from −1.0 to −1.5 μm, the total electron dose was 59.00 e$^-$/Å$^2$.

For the CXCR4-CXCL12 complex, the purified complex was obtained from a Superdex 6 Increase 10/300 GL column (Cytiva) and concentrated to 8 mg ml$^{-1}$. Additional CXCL12 was mixed with the CXCR4-CXCL12 complex at a 1.5:1 (CXCR4 calculated as monomer) molar ratio and incubated on ice for 2 h. The final concentration of the complex was adjusted to 6 mg ml$^{-1}$. FOM was added at a concentration of 0.5 mM, and the sample was applied onto glow-discharged UltrAuFoil 300 mesh R1.2/1.3 grids. Cryo-EM sample preparation followed the same protocol as before. Data were collected using a Titan Krios G2 transmission electron microscope (FEI) at 300 kV, equipped with a K3 direct detection camera (Gatan), at a pixel size of 0.826 Å at the New York Structural Biology Center (NYSBC). Movies were captured in super-resolution mode with a total electron dose of 52.80 e$^-$/Å$^2$.

For the CXCR4-gp120$^{HIV-2}$ complex, the peak fraction from a Superdex 6 Increase column was gathered and concentrated to 6.8 mg ml$^{-1}$, with 0.5 mM FOM added, using UltrAuFoil grids. Data collection took place at the Memorial Sloan-Kettering Cancer Center (MSKCC) on a Titan Krios G4 transmission electron microscope (FEI) at 300 kV, equipped with a Falcon 4i electron detector and managed with EPU software. The defocus was set between −0.8 and −2.2 μm, with a pixel size of 0.725 Å and a total electron dose of 60.21 e$^-$/Å$^2$.

For the CXCR4-gp120$^{HIV-2}$-CD4 complex, the peak fraction of the triple complex was collected and concentrated to 10 mg ml$^{-1}$, with 0.5 mM FOM incorporated into the sample. This mixture was applied to glow-discharged UltrAuFoil grids using the Vitrobot Mark IV, adhering to the same procedure as previously described. The blotting parameters were set to 2 s for time, 0 for force, and 10 s for wait time. The grids were screened at MSKCC, and data collection was conducted using the same parameters as those for the CXCR4-gp120$^{HIV-2}$ sample.

## Cryo-EM data processing
For human CXCR4 structure determination, a total of 4,043 movies were collected and processed using cryoSPARC[60], Patch motion correction and patch contrast transfer function (CTF) estimation were applied for drift correction and CTF parameter estimation, respectively. A total of 2,358,042 particles were blob-picked and extracted, then subjected to two rounds of 2D classification. The selected particles from the good classes were imported to generate an initial model, followed by heterogeneous refinement to improve map density. From the best class, 734,989 particles were selected for homogeneous refinement, followed by two rounds of non-uniform refinement with a symmetry change from C1 to C4, resulting in an EM map with an average resolution of 2.87 Å.

For the structure determination of the CXCR4-CXCL12 complex, 5807 movies were used for blob picking, resulting in 1,980,480 particles that were extracted and processed through two rounds of 2D classification. For the CXCL12-CXCR4 structure with an 8:8 stoichiometry, 229,609 particles were selected from the 2D classes that exhibited clear secondary structure features. These particles were used for ab-initio reconstruction and heterogeneous refinement, with 135,936 particles chosen from the best class for homogeneous and two non-uniform refinement steps with a symmetry transition from C1 to C4, leading to a final reconstruction with a 3.37 Å resolution. For the CXCL12-CXCR4 structure with an 8:4 stoichiometry, a similar process was followed, and 181,863 particles were used for non-uniform refinement under C4 symmetry, producing a map resolution of 3.46 Å.

For determining the structure of CXCR4-gp120$^{HIV-2}$, 11,031 movies were collected using the Falcon 4i camera. After particle selection through Blob picking and template-based methods, 2,400,728 particles were chosen for multiple rounds of 2D classification. After discarding junk particles, 1,080,238 particles were used for ab initio reconstruction and heterogeneous refinement to create a reliable initial model. The best model, which displayed gp120 binding, included 203,762 particles and was further refined using homogeneous and non-uniform methods, along with multiple 3D classification rounds.

This process produced four distinct models with slight conformational differences. Each model was individually refined using homogeneous and non-uniform techniques with bin2 to enhance map density. Specifically, 41,274 particles from class 2 were refined to 4.10 Å, 40,135 particles from class 3 were refined to 4.02 Å, 38,914 particles from class 4 were refined to 3.99 Å, and 38,098 particles from class 5 were refined to 4.10 Å. Particles from class 4 were re-extracted at bin1 and subjected to homogeneous refinement, followed by local refinement using a mask that included the CXCR4 tetramer and the V3 loop of gp120, excluding detergent micelles. This process yielded a 3.51 Å map, enabling precise modeling of the V3 loop structure.

For the CXCR4-gp120[HIV-2]-CD4 structure determination, 8007 movies with FOM were collected, and 1,219,168 particles were automatically extracted. These particles were processed through ab-initio reconstruction and heterogeneous refinement, resulting in an initial model showing both gp120 and CD4 bound. A further round of ab-initio and heterogeneous refinement was performed, and 51,034 particles were chosen for homogeneous and non-uniform refinement, producing a density map at 5.65 Å resolution. This map enabled the accurate placement of CXCR4, along with docking gp120[HIV-2] and CD4. All reported maps were generated using gold-standard refinement procedures, with the FSC (Fourier Shell Correlation) cutoff set at 0.143.

### Model building, structure refinement, and visualization

The models for CXCR4 tetramer, CXCR4-CXCL12, and CXCR4-gp120[HIV-2] were predicted using Alphafold2[61], while the CD4 model was derived from the published CCR5-gp120[HIV-1]-CD4 structure (PDB:6MEO)[21]. The N-terminal domain of CXCR4 and the V3 loop of gp120[HIV-2] were constructed de novo. Structure modeling was conducted with Coot[62], and atomic coordinates were refined against the map through real-space refinement in PHENIX[63]. Visualization and figure generation were performed using Chimera[64] and ChimeraX[65]. Adobe Photoshop was utilized for the layout of the images.

### Pull-down assay

To verify the interactions between CXCR4 and gp120[HIV-2], we performed an affinity pull-down assay using Flag-tagged CXCR4. To do this, genes of wild-type and mutant gp120[HIV-2] (residues 23–501) were fused with c-Myc-tag at their C terminus, and then cloned into pcDNA3.4 vector separately. In brief, 50 ml Expi293 cells at a density of $3.0 \times 10^6$ ml⁻¹ were transfected using 40-kDa linear PEI for CXCR4 expression. Cells were collected 4 days after transfection. The purification was performed similarly, as described above, using LMNG and Flag resin. The resin was washed by the wash buffer III (25 mM HEPES, pH 7.5, 150 mM NaCl, 0.01 % (w/v) LMNG, and 0.001% (w/v) CHS) and then incubated with gp120[HIV-2]. For secreted expression of gp120[HIV-2], 30 ml Expi293 cells at a density of $3.0 \times 10^6$ ml⁻¹ were transfected with plasmids and PEI at a ratio of 1:2.5. Supernatant from cell culture was collected 7 days after transfection, followed by dialysis against 100× excess of dialysis buffer for 48 h at 4 °C. Then, CXCR4-bound Flag resin was incubated with wild-type and mutant gp120[HIV-2] supernatant (including 0.01% LMNG and 0.001% CHS) for 2 h at 4 °C. The resin was extensively washed using 60 column volumes of wash buffer III and eluted with wash buffer III supplemented with 0.2 mg ml⁻¹ 3×Flag peptide. Eluted proteins were subjected to SD-PAGE. Proteins were electro-transferred onto PVDF membranes using a semi-dry transfer cell (Bio-Rad). PVDF membranes were blocked using 5% (w/v) nonfat dry milk (Bio-Rad) for 1 h at room temperature. Following incubation with horseradish peroxidase (HRP)-conjugated anti-Flag-tag (86861S, Cell Signaling) or anti-Myc-tag antibodies (2040S, Cell Signaling) (1:2000-fold dilution (v/v)) overnight at 4 °C, PVDF membranes were washed in 0.05% Tween-20/TBS. The bound antibodies were visualized by using an enhanced chemiluminescence reagent (Thermo Scientific) and quantified by densitometry using a ChemiDoc MP imaging system (Bio-Rad). The assays were repeated twice with consistent results, and the source data are provided in the Source data file.

### Co-immunoprecipitation (Co-IP) assay

Wild-type and mutant CXCL12 were coexpressed with CXCR4 due to their poor secreted expression. A total of 35 µg plasmids encoding CXCR4 and CXCL12 were mixed with 88 µg PEI. The mixture was subsequently added to a suspension of Expi293 cells when the cell density reached $3.0 \times 10^6$ cells ml⁻¹. Cells were cultured for 5 days before harvest. The purification was performed similarly to above using Flag resin. The resin was washed thoroughly using wash buffer III before elution. The eluted proteins were analyzed by western blot experiments using HRP-conjugated antibodies against Flag-tag and Myc-tag with a 1:2000-fold dilution (v/v) to detect CXCR4 and CXCL12, respectively. The bound antibodies were visualized by using an enhanced chemiluminescence reagent and quantified by densitometry using a ChemiDoc MP imaging system (Bio-Rad). The assays were repeated twice with consistent results, and the source data are provided in the Source data file.

### Native PAGE

The expression and purification of CXCR4, CXCL12-CXCR4, HIV-2 gp120, and CD4-ECD were the same as described above, except that eluted CXCR4 was subjected to SEC using a Superdex 200 Increase 10/300 GL column (GE Healthcare). Moreover, CXCR4 (calculated as monomer) was assembled with HIV-2 gp120 or gp120-CD4 at a 1:6 molar ratio to guarantee the maximum assembly efficiency. Purified CXCR4, CXCL12-CXCR4, CXCR4-gp120, and CXCR4-gp120-CD4 proteins (total 3 µg each) were loaded into native PAGE (4 to 16%, Thermo Scientific). The gel was run at low temperature (4 °C) with 150 V Constant for 60 min and then 250 V Constant for another 60 min. The gel was fixed in the solution (40% methanol, 10% acetic acid) and heated for 45 s. After shaking for 15 min at room temperature, destain solution (8% acetic acid) was added and the gel was heated for 45 s. The bands were recorded using the Gel Doc XR+ imaging system (Bio-Rad) when the desired background is obtained. Uncropped gels are provided in the Source data file.

### Mass photometry

Mass photometry experiments were performed using a Refeyn TwoMP instrument. A pre-assembled 6-well sample cassette (Refeyn) was placed at the center of a clean sample carrier slide (Refeyn), with each well designated for an individual measurement. To establish the focal point, 15 µL of freshly prepared SEC buffer (25 mM HEPES pH 7.5, 150 mM NaCl, 0.001% LMNG, 0.0001% CHS, and 0.00033% GDN) was added to the well. The focus was determined and maintained throughout the measurement using an autofocus system based on total internal reflection. Purified CXCR4, CXCR4-gp120[HIV-2], and CXCR4-gp120[HIV-2]-CD4 proteins, prepared in the same SEC buffer, were initially diluted to 200 nM, and 3 µL of the diluted sample was added to the buffer drop, achieving a final protein concentration of 33.3 nM. Once the autofocus stabilized, movies were recorded for 60 s. Data were collected using Refeyn AcquireMP (version 2024.1.1.0) and analyzed with Refeyn DiscoverMP (version 2024.1.0.0). Contrast-to-mass calibration was carried out using a BSA protein standard (Sigma), which contained BSA monomers and dimers with molecular masses of 66.5 and 132 kDa, respectively. Statistical analysis was conducted using DiscoverMP, where Gaussian fitting was applied to distribution peaks to determine the average molecular mass of each component. Plotting was carried out using GraphPad Prism 10. Source data are provided in the Source data file.

### Deglycosylation assay

PNGase F (P0704S, NEB) was used for removing *N*-linked oligosaccharides from CXCR. Briefly, 0.42 µg CXCR4 proteins (tetramer or monomer) were mixed with 1× GlycoBuffer 2 and 0.5 µl PNGase F to a

total reaction volume of 20 μL. Reaction without PNGase F served as negative controls. After incubation at 37 °C for 1.5 h, the separation of reaction products was analyzed by western blot experiments using HRP-conjugated antibodies against Flag-tag (1:2000-fold dilution (v/v)). The bound antibodies were visualized by using an enhanced chemiluminescence reagent and quantified by densitometry using a ChemiDoc MP imaging system (Bio-Rad). An uncropped and unprocessed blot is provided in the Source data file.

## Reporting summary

Further information on research design is available in the Nature Portfolio Reporting Summary linked to this article.

## Data availability

All data supporting the conclusions of this study can be obtained from the corresponding author upon request. The atomic coordinates and cryo-EM density map have been deposited in the PDB and EMDB database under the ID codes PDB: 9MDU and EMDB: EMD-48180 (CXCR4 tetramer); PDB: 9ME1 and EMDB: EMD-48182 (CXCL12-CXCR4 complex with 8:8 stoichiometries); PDB: 9MEU and EMDB: EMD-48220 (CXCL12-CXCR4 complex with 8:4 stoichiometries), PDB: 9MEJ and EMDB: EMD-48215 (global CXCR4-gp120$^{HIV-2}$ complex), PDB: 9MEN and EMDB: EMD-48218 (CXCR4-gp120$^{HIV-2}$/V3 loop complex) and PDB: 9MET and EMDB: EMD-48219 (CXCR4-gp120$^{HIV-2}$-CD4 complex). PDB entries (1A15, 1QG7, 2K01, 3OE9, 3OE0, 4RWS, 5VN3, 6MEO, 8K3Z, 8U4P, 8U4T, 8YU7) used in this study were downloaded from the Protein Data Bank. The source data underlying Figs. 2 and 3, and Supplementary Figs. 1, 3 and 5 are provided as a Source data file. Source data are provided with this paper.

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

## Acknowledgements

This research was supported by funds from the Maloris Foundation and Memorial Sloan-Kettering Cancer Center Core grant P30CA008748 to D.J.P. The authors thank access to cryo-EM facilities at the National Center for CryoEM Access and Training (NCCAT) and Simons Electron Microscopy Center at the New York Structural Biology Center, supported by the NIH Common Fund Transformative High Resolution Cryo-Electron Microscopy program (U24GM129539) and by grants from the Simons Foundation (SF349247) and NY State Assembly. L.Z. was supported by NIH HD110120, awarded to Scott Keeney and D.J.P. The authors thank members of the Patel and Xiao laboratories for discussions and experimental advice. The authors thank J. De La Cruz (MSK), the staff at the Simons Electron Microscopy Center, as well as the Peking University Cryo-EM facility and high-performance computing platforms for their support in cryo-EM data collection.

## Author contributions

Z.Z., J.X., and D.J.P. conceived the study. Z.Z and H.Z. optimized and purified the proteins with assistance from S.C. and D.S. Z.Z. performed cryo-EM sample preparation, data collection, and data processing. Z.Z. and L.Z. carried out model building, data analysis, and mutation analysis for the biochemical assays under the guidance of D.J.P. H.Z. conducted the biochemical assays. Z.Z., L.Z., and D.J.P. drafted the manuscript with input from H.Z. J.X. revised the draft, and the final manuscript was approved by all authors.

## Competing interests

The authors declare no competing interests.
