## [Peer Review file · Nature Communications]

CXCR4 chemokine receptor : HIV envelope spike recognition and cognate CXCL12 inhibition

Corresponding Author: Professor Dinshaw Patel

Version 0:

Reviewer comments:

Reviewer #1

(Remarks to the Author)

CXCR4 plays a critical role in immune regulation and HIV recognition, serving as a co-receptor for certain HIV strains. Different HIV strains prefer either CCR5 or CXCR4 for viral entry, yet the underlying molecular mechanisms remain elusive. In the current study, Zhang et al. report the structures of CXCR4 alone and the CXCL12-CXCR4, CXCR4-gp120 and CXCR4-gp120-CD4 complexes, of which CXCR4 adopts an unanticipated tetrameric state. The structures reveal that one or two gp120 (HIV-2) molecules bind to the CXCR4 tetramer via their V3 loops, which insert into the major sub-pocket of the receptor. Additionally, the N terminus of CXCL12 would clash with the V3 loop of gp120 upon binding to CXCR4, elucidating the structural basis of CXCL12 blocking the HIV entry. These insights open new avenues for developing the anti-HIV strategies.

Overall, this study for the first time elucidates the molecular interaction between CXCR4 and HIV-2 gp120, a discovery with significant implications. Nonetheless, several specific issues require further clarification before publication.

1. Only single peak in gel filtration of CXCR4 was observed (Extended Data Fig. 1), whereas the previous work by Kei Saotome et al. (Nat Struct Mol Biol, 2024) detected a mixture of monomeric and oligomeric forms of the receptor. It is therefore recommended to include a native PAGE analysis to better assess the sample's oligomeric state. Additionally, the cryo-EM data should be re-examined to determine whether monomeric or alternative oligomeric forms of CXCR4 are present.

Furthermore, the retention volumes for CXCR4 alone and the CXCL12-CXCR4 complex appear to be identical (Extended Data Fig. 1a and 1b). However, the final cryo-EM reconstruction reveals similar particle numbers for the 2:1 and 1:1 stoichiometries of the CXCL12-CXCR4 complex. This discrepancy suggests that the addition of FOM during cryo-sample preparation might induce the formation of the dimeric tetramer. The oligomeric states of the CXCL12-CXCR4 complex should be further assessed.

2. The authors report that the sample of CXCR4-gp120HIV-2 complex was prepared at a 1:2 molar ratio of CXCR4 and gp120. And the resolved structure shows that CXCR4 binds only one gp120. It would be worth investigating whether increasing the gp120 ratio would produce a full complement of gp120 molecules (e.g., three or four). This should also be explored for the CXCR4-gp120-CD4 complex, which was prepared at a 1:1.3 molar ratio as described in the methods.

3. In the Methods section, please provide detailed information regarding the specific HIV strains used for the gp120 protein preparation.

4. In the cryo-EM sample preparation and data collection section, the authors report using Quantifoil Au 300 mesh R1.2/1.3 for the CXCR4 apo sample, which conflicts with an earlier description that mentioned holey-gold UltrAufoil grids in results section. This discrepancy should be double-checked for consistency. Additionally, the data processing section should specify the symmetry imposed during the final refinement for each dataset. For the CXCR4-gp120HIV-2 sample, the flowchart shows that particles were refined using both bin2 and bin1. This detail should be explicitly stated in the methods. Finally, the description of the local refinement for the V3 loop is unclear. Including a masked map that highlights the focused region in the cryo-EM processing flowchart would greatly improve clarity.

Reviewer #2

(Remarks to the Author)

In this manuscript, the authors presented several CXCR4 structures including homo-tetramers without ligand or G protein, CXCR4 tetramers bound to ligand (CXCL12 monomer or dimer), and CXCR4 bound to HIV-2 env gp120. A characteristic feature of this work is the CXCR4 tetramer, which is inactive (not coupled to G proteins) and able to transit to monomers with

biological functions. The bindings of CXCL12 dimers and the HIV-2 gp120 further extend the manuscript that advances the research field with solid structural evidence. There are, however, a number of concerns that should be addressed by the authors.

Specific comments:

- 1, All CXCR4 structures obtained are tetramers, whereas in other published reports monomers (Gi protein-coupled), dimers and trimers are found. Sample preparation methods should be compared and any differences identified. Did the inclusion of a C terminal FLAG tag play a role in tetramer formation?
- 2, The authors found that the N terminus of CXCR4 was flexible and a cryo-EM structure could not be obtained. This is in contrast to general believe that the N terminus of the chemokine receptor wraps around the core structure of the chemokine. Could the observed phenomenon a feature specific to the tetrameric CXCR4?
- 3, No functional assays were included in this manuscript. Did the R12A mutation affect function of CXCL12 in activation of CXCR4?
- 4, The binding of CXCL12, whether in monomer or dimer, induces transmembrane signaling through CXCR4. Signaling assays are extensively used in structural studies to test and confirm agonist binding, often with mutations of key binding residues. It is suggested that authors include CXCR4-based signaling assays with receptor mutants to confirm their models. This is especially important in consideration that CXCL12 dimer activates CXCR4 in a different way from CXCL12 monomer, as reported in Ref# 29. The binding to vs. activation of CXCR4 by both CXCL12 monomer and dimer should be examined using approaches reported in that reference, in order to find any structural difference underlying the signaling difference.
- 5, Is there experimental proof that CXCR4 tetramers exist in intact cells?
- 6, How did CXCL12a and CXCL12b interact with each other? Providing additional details of the molecular interaction could be helpful.

Minor comments:

- 1, The authors are suggested to compare their tetramer structure with recently reported CXCR4 tetramer from other groups (Saotome et al. 2025, PMID 39313635 ; Liu et al. 2025, PMID 39891908) and comment on similarities and differences if any.
- 2, While theoretically 4 HIV-2 gp120 env may be bound to the CXCR4 tetramer, the authors observed only one gp120 binding. Could this be the result of steric clash as the gp120 protein is quite large?
- 3, The "sandwich" model in which CXCL12 dimers bind CXCR4 on both sides was interesting and could potentially serve a role for juxtacrine signaling, but is there any experimental evidence for its presence?
- 4, Page 6, second paragraph: "This shift alters the configuration of CXCL12, particularly the N-terminal loop": should be "conformation" instead of "configuration".
- 5, Fig 4d: HIV-2's slower progression and lower transmissibility compared to HIV-1 hint at Env differences. Its gp120 may have distinct binding dynamics with CXCR4, potentially influencing entry dynamics. Authors may want to discuss on this to elaborate their modeled findings, and this discussion would provide more comprehensive implications of their study for understanding the distinctive properties of HIV-2 Env interacting with CXCR4.
- 6, Given the relevance of CXCR4 antagonists for HIV therapeutics, Extended Data Fig. 9 should be moved to the main figure part.

Reviewer #3

(Remarks to the Author)

CXCR4 is a chemokine receptor that interacts with its ligand CXCL12 and plays an important role in cell migration and hematopoiesis, and it also serves as one of the HIV co-receptors. High-resolution structural information would be critical for our understanding of the molecular mechanisms of how CXCR4 mediates both cellular regulation and HIV inhibition. In this manuscript, Zhang et al. reported cryo-EM structures of CXCR4 alone, CXCR4-CXCL12 complex, CXCR4-gp120 complex and CXCR4-gp120-4D CD4 complex. In all cases, CXCR4 unexpectedly formed a tetramer, different from what have been reported previously. CXCL12 bound to CXCR4 in the same way as other chemokines bind to their receptor. The binding mode of HIV-2 gp120 to CXCR4 is also very similar to that of how HIV-1 gp120 binds to CCR5. The authors suggest that their "study provides fundamental structure-based mechanistic insights into how HIV-2 gp120 binds its co-receptor CXCR4 and explains how CXCL12 blocks HIV entry, offering guidance for the development of vaccines and therapeutic agents targeting HIV infection".

Structural studies of CXCR4 and its complexes with various ligands are highly significant and should be of great interest to readers in both the GPCR and HIV fields. There are some issues with the current version of manuscript, however.

Major points:

1. While the density for CXCR4 is quite good in most of the structures except for the CXCR4-gp120-CD4 complex, the density for gp120 is rather poor (at least based on all the figures presented), even for the V3 loop that inserts into the chemokine-binding pocket (CRS2) and is supposed to be rigid. It is unclear whether their model for gp120 binding to CXCR4 would be more reliable than an AlphaFold-predicted model. It would be a good idea to tone down the discussion about the interactions at the atomic level.
2. The HIV-2 gp120:CXCR4 ratio looks very odd by SDS-PAGE in Extended Data Fig. 1.c and d, with too much gp120 and almost no CXCR4, inconsistent with the stoichiometry revealed by the structures. In the meanwhile, Extended Data Fig. 10 showed a reasonable ratio for HIV-1 gp120 and CXCR4, even though the complex is supposed to have lower binding affinity.
3. These structures have not provided answers to some important questions in the HIV field, such as, what are structural

determinants for co-receptor usage (R5 vs X4, or R5X4), and how the virus manages to switch coreceptor. Again, it may have to do with the quality of the cryo-EM maps that could not provide enough information at high resolution.

Minor points:

1. Writing of the manuscript needs to be improved. The authors should be mindful of all the terms they use. For example, in the sentence -“The subsequent conformational changes in the fusion peptide gp41 bring the viral and host membranes together, promoting their fusion”. Most likely, they meant “the fusion subunit gp41”. Fusion peptide means a completely different thing in the viral entry field. Also, not sure why the CXCR4 tetramer is called CXCR4 complex? Complex with what?
2. It was first stated that “The N-terminus of CXCR4 acts as the chemokine recognition site 1 (CRS1), engaging with the globular core of CXCL12”, and then “Similar to the CXCR4 tetramer complex alone, the N-terminal residues (1-25) of CXCR4 in the CXCL12-CXCR4 complex are not visible, likely due to inherent flexibility.” One of the statements has to be incorrect.
3. “Notably, the monomer monomer interfaces in our cryo-EM structures and those in the crystal structures are strikingly different (Fig. 1b). Recent studies have also shown that CXCR4 can exist in monomeric and trimeric forms 25,28, highlighting the highly dynamic nature of CXCR4 assembly. This structural flexibility may be linked to its functional diversity, as different CXCR4 conformations could correspond to distinct cellular outcomes.” Could it be caused by different conditions used for different studies?
4. “Our structures offer valuable insights for the development of vaccines and therapeutic agents targeting both HIV-1 and HIV-2 infections. One promising strategy is to design neutralizing antibodies that target the V3 loop of HIV-1 and HIV-2, focusing specifically on the GPGR motif in HIV-1 and GFKF motif in HIV-2”. The authors need to get themselves more familiar with the HIV literature (or at least the manuscript should be looked at by an HIV expert). There are many antibodies against the V3 crown (GPGR/Q) and they are not useful for vaccine development.
5. “gp120HIV-2 (residues 23-501) were fused”. Which HIV-2 strain was used? Is it a full-length gp120?
6. Representative density for gp120 should be shown in Extended Data Fig. 5. Density for gp120 is very bad in Extended Data Fig. 8.

Version 1:

Reviewer comments:

Reviewer #1

(Remarks to the Author)

The authors have addressed all my concerns.

Reviewer #2

(Remarks to the Author)

The authors made a great effort to address all raised comments, resulting in a much improved manuscript. While the lack of functional validation with mutagenized CXCR4 remains a concern, the structural information contained in the manuscript provides important advancement to the research field. There is no additional comments from this reviewer.

Reviewer #3

(Remarks to the Author)

The authors have addressed my previous comments.

Response to reviewers

We are very grateful to the reviewers for their insightful and constructive comments. Their feedback has helped us refine and strengthen the manuscript, and we believe the paper has significantly improved as a result. Below we provide a point-by-point response to their concerns.

Reviewer #1

CXCR4 plays a critical role in immune regulation and HIV recognition, serving as a co-receptor for certain HIV strains. Different HIV strains prefer either CCR5 or CXCR4 for viral entry, yet the underlying molecular mechanisms remain elusive. In the current study, Zhang et al. report the structures of CXCR4 alone and the CXCL12-CXCR4, CXCR4-gp120 and CXCR4-gp120-CD4 complexes, of which CXCR4 adopts an unanticipated tetrameric state. The structures reveal that one or two gp120 (HIV-2) molecules bind to the CXCR4 tetramer via their V3 loops, which insert into the major sub-pocket of the receptor. Additionally, the N terminus of CXCL12 would clash with the V3 loop of gp120 upon binding to CXCR4, elucidating the structural basis of CXCL12 blocking the HIV entry. These insights open new avenues for developing the anti-HIV strategies.

Overall, this study for the first time elucidates the molecular interaction between CXCR4 and HIV-2 gp120, a discovery with significant implications. Nonetheless, several specific issues require further clarification before publication.

We thank the reviewer for their thoughtful and encouraging comments on our study. We appreciate the positive evaluation and have addressed all points raised in detail below. Where needed, we have added new data and clarifications to strengthen our conclusions.

1. Only single peak in gel filtration of CXCR4 was observed (Extended Data Fig. 1), whereas the previous work by Kei Saotome et al. (*Nat Struct Mol Biol*, 2024) detected a mixture of monomeric and oligomeric forms of the receptor. It is therefore recommended to include a native PAGE analysis to better assess the sample's oligomeric state. Additionally, the cryo-EM data should be re-examined to determine whether monomeric or alternative oligomeric forms of CXCR4 are present.

Thank you for your kind reminder. In the paper by Kei Saotome et al. (*Nat Struct Mol Biol*, 2024), a tandem column setup was used to improve the separation of different CXCR4 species, specifically, a Superose 6 Increase 10/300 GL column connected upstream of a Superdex 200 Increase 10/300 GL column. In contrast, for the preparation of our cryo-EM samples, we used only a Superose 6 Increase 10/300 GL column. To evaluate whether different oligomeric states of CXCR4 were present in our purified samples, we repeated the purification following the method used in their study.

Our results showed that the main peak corresponded to the CXCR4 tetramer, as confirmed by SDS-PAGE and BN-PAGE (**Supplementary Fig. 1, panels a, c and e**). A smaller peak appeared after the main peak, and the fractions collected from it displayed the same bands as the tetramer on SDS-PAGE. Native PAGE further confirmed that this smaller peak corresponded to the CXCR4 monomer, which exhibited a significantly lower molecular weight, approximately one-fourth that of the tetrameric CXCR4 (~120 kDa vs. ~480 kDa). (**Supplementary Fig. 1, panels b-e**). We also noted that the CXCR4 bands were somewhat diffuse, which we suspected was due to glycosylation. Treatment with PNGase F, an enzyme that removes most N-linked oligosaccharides from glycoproteins, caused both the tetramer and monomer bands to shift to

lower molecular weights and become more defined (**Supplementary Fig. 1, panel f**). This confirmed the presence of both CXCR4 monomer and tetramer in our sample and suggested that they were likely N-linked glycosylated. Additionally, 2D classification of our cryo-EM dataset revealed that approximately 3% of the particles (8,035 out of 182,049) corresponded to the monomeric state (**Supplementary Fig. 2, panel d**), supporting its presence in the sample.

Several factors can influence the oligomeric states of membrane proteins, including the expression system, detergent type, critical micelle concentration (CMC) of the detergents, and the separation efficiency of the SEC column. We carefully analyzed the differences in CXCR4 sample preparation between previous studies and our own, considering factors such as the construct used, expression system, detergent type, and detergent concentration (**Response Table 1**, see below). In our apo samples, we observed monomeric and tetrameric forms of CXCR4, but found no clear evidence of dimeric or trimeric states. These differences may reflect the impact of the lipid environment associated with the expression system (e.g., insect cells vs. mammalian cells) on CXCR4 oligomerization (PMID: 39313635; PMID: 39891908). The detailed methods and results, as well as **Supplementary Fig. 1** and **Supplementary Fig. 2, panel d** have been updated in our revised manuscript.

Results:

“Overall structure of human CXCR4 tetramer

The elution volume of gel filtration suggested that CXCR4 primarily exists as an oligomer. Additionally, we observed a minor peak following the main peak. SDS-PAGE analysis confirmed that both peaks correspond to CXCR4 protein (**Supplementary Figs. 1a-c**). Native page showed that the minor peak has an apparent molecular weight of approximately 120 kDa, while the main peak is around 480 kDa- roughly four times larger. Based on these observations and consistent with a recent publication²⁵, we assigned these two peaks to monomeric and tetrameric forms of CXCR4, respectively (**Supplementary Figs. 1d, e**). The rising molecular weights observed on the native gel are likely influenced by detergent micelles surrounding the CXCR4 proteins.

On the SDS page, we observed diffuse CXCR4 bands, which shifted to a low molecular weighted upon treatment with PNGase F-an enzyme for removing almost all N-linked oligosaccharides from glycoproteins, suggesting that CXCR4 is highly glycosylated (**Supplementary Fig. 1f**).

2D classification identified a minor population (~3%) of monomeric CXCR4 (**Supplementary Fig. 2d**). However, the limited particle number and size constraints of cryo-EM prevented us from reconstructing a high-resolution structure.”

Methods:

“Native PAGE

The expression and purification of CXCR4, CXCL12-CXCR4, HIV-2 gp120 and CD4-ECD were same as described above except that eluted CXCR4 was subjected to size-exclusion chromatography (SEC) using a Superdex 200 Increase 10/300 GL column (GE Healthcare). Moreover, CXCR4 (calculated as monomer) was assembled with HIV-2 gp120 or gp120-CD4

at a 1:6 molar ratio to guarantee the maximum assembly efficiency. Purified CXCR4, CXCL12-CXCR4, CXCR4-gp120 and CXCR4-gp120-CD4 proteins (totally 3 μ g each) were loaded into native PAGE (4 to 16%, Thermo Scientific). The gel was run in low temperature (4°C) with 150 V Constant for 60 mins and then 250 V Constant for another 60 mins. The gel was fixed in the solution (40% methanol, 10% acetic acid) and heated for 45 seconds. After shaking for 15 minutes at room temperature, destain solution (8% acetic acid) was added and the gel was heated for 45 seconds. The bands were recorded by using Gel Doc XR+ imaging system (Bio-Rad) when the desired background is obtained.

Deglycosylation assay

PNGase F (P0704S, NEB) was used for removing *N*-linked oligosaccharides from CXCR4. Briefly, 0.42 μ g CXCR4 proteins (tetramer or monomer) were mixed with 1 \times GlycoBuffer 2 and 0.5 μ l PNGase F to a total reaction volume of 20 μ l. Reaction without PNGase F served as negative controls. After incubation at 37°C for 1.5 hour, the separation of reaction products was analyzed by western blot experiments using HRP-conjugated antibodies against Flag-tag (1:2,000-fold dilution (v/v)). The bound antibodies were visualized by using an enhanced chemiluminescence reagent and quantified by densitometry using a ChemiDoc MP imaging system (Bio-Rad)."

Supplementary Fig.1

a Size exclusion chromatography (SEC) of purified CXCR4 expressed in mammalian cells showing distinct peaks corresponding to aggregates, tetramers, monomers, and free Flag peptides.

b,c SDS-PAGE analysis confirms the monomeric (b) and tetrameric (c) CXCR4 bands related to a.

d,e Blue native PAGE (BN-PAGE) analysis showing distinct bands for CXCR4 monomer (d) and tetramer (e).

f Western blot analysis with and without PNGase F treatment confirms glycosylation of CXCR4 in both monomeric and tetrameric forms.

Supplementary Fig.2, panel d:

d. Class averages showing a minor population of monomeric CXCR4 particles. From 182,049 total particles in this subset, 8,035 particles (~3%) were classified as monomeric (highlighted in red boxes).

Response Table 1: “**Comparison of CXCR4 Conformational States**”

Oligomeric states	Structure /complex	Expression strategy	Expression system	Detergent for SEC	Oligomeric analysis
Monomer	CXCL12-CXCR4-Gi-scFv16	Co-expression of N-Flag-BRIL-CXCR4, CXCL12-8xHis-C, G α i, G β 1, and G γ 2; followed by incubation with CXCL12 and scFv16 and SEC purification	Insect cells	0.00075% LMNG, 0.0002% CHS	Cryo-EM (PMID:39093700) 8K3Z
	CXCR4_{EM}-Gi-CXCL12	Co-expression of CXCR4-monomeric eGFP-Flag-C, His-tagged G α i, and His-tagged G β 1-G γ 2; incubated with CXCL12 prior to grid preparation	Insect cells	0.01% LMNG, 0.001% CHS	Cryo-EM (PMID:39313635) 8U4O
	HF51116/AMD070/AMD3100 - CXCR4^{κOR}-Nb6	Purified N-Flag-BRIL-CXCR4 (L125W, T240P; ICL3 replaced by κ OR segment)-10xHis-C with antagonist; incubated with Nb6-8xHis-C and purified by SEC	Insect cells	0.01% LMNG, 0.002% CHS	Cryo-EM (PMID:40063796) 8ZPL 8ZPM 8ZPN
	CXCR4	CXCR4- monomeric eGFP-Flag-C	Insect cells	0.01% LMNG, 0.001% CHS	SEC and Native-PAGE (PMID:39313635)
	CXCR4	CXCR4-Flag-C	Expi293 cells	0.01% LMNG, 0.001% CHS, 0.00033% GDN,0.5mM FOM	SEC,Native-PAGE and Mass photometry (this study)
Dimer	CXCR4-ITIt/CVX15	Purified N-Flag-3C-CXCR4-10xHis-C in the presence of antagonist	Insect cells	0.05%DDM, 0.01% CHS	Crystal structure (PMID:20929726) 3ODU 3OE0
Trimer	CXCR4_{EM}-REGN7663 Fab	CXCR4-monomeric eGFP-Flag-C mixed with REGN7663 Fab prior to grid preparation	Insect cells	0.01% LMNG, 0.001% CHS	Cryo-EM (PMID:39313635) 8U4S 8U4T
	CXCR4_{EM}-REGN7663 Fab				
Tetramer	CXCR4	N-Flag-CXCR4	Insect cells	0.00075% LMNG, 0.0002% CHS	Cryo-EM (PMID:39891908) 8YU7
	CXCR4	CXCR4-Flag-C	Expi293 cells	0.01% LMNG, 0.001% CHS, 0.00033% GDN, 0.5 mM FOM	Cryo-EM (this study)

Furthermore, the retention volumes for CXCR4 alone and the CXCL12-CXCR4 complex appear to be identical (Extended Data Fig. 1a and 1b). However, the final cryo-EM reconstruction reveals similar particle numbers for the 2:1 and 1:1 stoichiometries of the CXCL12-CXCR4 complex. This discrepancy suggests that the addition of FOM during cryo-sample preparation might induce the formation of the dimeric tetramer. The oligomeric states of the CXCL12-CXCR4 complex should be further assessed.

We corrected the description of the CXCL12:CXCR4 stoichiometry to 8:4 and 8:8, instead of 2:1 and 1:1, for clarity. To further investigate the stoichiometry of the CXCL12-CXCR4 complex in the absence of FOM, we performed native PAGE analysis. The gel showed that the molecular weight of the CXCR4 tetramer closely matched that of the major CXCL12-CXCR4 complex population, consistent with a 8:4 stoichiometry (CXCL12:CXCR4) (**Supplementary Fig. 3e**). Interestingly, we also observed an additional band above the main complex, which may correspond to a head-to-head CXCL12-CXCR4 assembly with an 8:8 stoichiometry (**Supplementary Fig. 3e**). Moreover, 2D classification of the CXCL12-CXCR4 complex (without FOM) revealed a substantial population of dimeric structures. Although the functional relevance of this dimeric form remains unclear, we believe it is important to include this observation in the main text for completeness.

Results:

“Two distinct assembly mechanisms of CXCL12-CXCR4 complex formation

To understand how the cognate ligand CXCL12 binds to its receptor CXCR4 and the mechanism by which it blocks HIV entry, we co-expressed CXCL12 and CXCR4 that yielded a single peak on a size-exclusion column (**Supplementary Fig. 3a**) and determined the structure of the complex. We obtained two distinct structures of the CXCL12-CXCR4 complex (**Supplementary Figs. 3b-d**), namely with 8:4 stoichiometry at 3.4 Å (**Fig. 2a**) and 8:8 stoichiometry at 3.3 Å (**Fig. 2b**). Both states were confirmed by native PAGE analysis (**Supplementary Fig. 3e**).”

Methods:

“Native PAGE

The expression and purification of CXCR4, CXCL12-CXCR4, HIV-2 gp120 and CD4-ECD were same as described above except that eluted CXCR4 was subjected to size-exclusion chromatography (SEC) using a Superdex 200 Increase 10/300 GL column (GE Healthcare). Moreover, CXCR4 (calculated as monomer) was assembled with HIV-2 gp120 or gp120-CD4 at a 1:6 molar ratio to guarantee the maximum assembly efficiency. Purified CXCR4, CXCL12-CXCR4, CXCR4-gp120 and CXCR4-gp120-CD4 proteins (totally 3µg each) were loaded into native PAGE (4 to 16%, Thermo Scientific). The gel was run in low temperature (4°C) with 150 V Constant for 60 mins and then 250 V Constant for another 60 mins. The gel was fixed in the solution (40% methanol, 10% acetic acid) and heated for 45 seconds. After shaking for 15 minutes at room temperature, destain solution (8% acetic acid) was added and the gel was heated for 45 seconds. The bands were recorded by using Gel Doc XR+ imaging system (Bio-Rad) when the desired background is obtained.”

Supplementary Fig.3 panel e:

e BN-PAGE analysis of CXCL12-CXCR4 complexes reveals the formation of both 8:4 and 8:8 stoichiometries when assembled at a 1:6 molar ratio (monomer basis).

Extra figure for reviewer's consideration:

2D classification for CXCR4-CXCL12 sample without FOM

2D classification of the CXCL12-CXCR4 complex in the absence of FOM. Selected 2D class averages are shown. Particles highlighted with red boxes correspond to the dimeric tetramer form of the CXCL12-CXCR4 complex (8:8 stoichiometry).

2. The authors report that the sample of CXCR4-gp120^{HIV-2} complex was prepared at a 1:2 molar ratio of CXCR4 and gp120. And the resolved structure shows that CXCR4 binds only one gp120. It would be worth investigating whether increasing the gp120 ratio would produce a full complement of gp120 molecules (e.g., three or four). This should also be explored for the CXCR4-gp120-CD4 complex, which was prepared at a 1:1.3 molar ratio as described in the methods.

We appreciate your valuable comments. To probe complex formation, we assembled the CXCR4-gp120^{HIV-2} and CXCR4-gp120^{HIV-2}-CD4 complexes at a 1:6 molar ratio (i.e., an excess of gp120 or gp120^{HIV-2}-CD4, corresponding to three gp120^{HIV-2} or gp120^{HIV-2}-CD4 complexes per CXCR4 monomer). The molecular weights of the resulting complexes were analyzed using native PAGE and mass photometry. Mass photometry revealed that the CXCR4-gp120^{HIV-2} complex exhibited a MW increase of approximately 93 kDa compared to the CXCR4 tetramer alone, suggesting the binding of one or two gp120 molecules (each ~58.2 kDa) to the CXCR4 tetramer (**Supplementary Fig. 5, panels e and f**). For the CXCR4-gp120^{HIV-2}-CD4 complex, the MW increase was ~145 kDa, while the gp120^{HIV-2}-CD4 subcomplex has a theoretical MW of ~102 kDa (**Supplementary Fig. 5, panels e and g**). This suggests the presence of one or two gp120^{HIV-2}-CD4 subcomplexes

bound to the CXCR4 tetramer, consistent with our cryo-EM data, which show either one or two gp120 molecules bound to the CXCR4-gp120^{HIV-2} complex, and two gp120^{HIV-2}-CD4 molecules bound within the CXCR4-gp120^{HIV-2}-CD4 complex. Additionally, native PAGE showed corresponding band shifts for these complexes (**Supplementary Fig. 5, panels c and d**). These results have been updated in the main text, Methods section, and **Supplementary Fig. 5, panels c-g**.

Results:

“Reconstitution of CXCR4-gp120^{HIV-2}-CD4 complex

Native page gel results showed a broad distribution of CXCR4-gp120^{HIV-2} and CXCR4-gp120^{HIV-2}-CD4 complexes (**Supplementary Figs. 5c, d**), making it is difficult to determine the exact stoichiometry of gp120 and CD4 binding to CXCR4 tetramer. To further clarify this, we employed mass photometry to assess their molecular weights (**Supplementary Figs. 5e-g**). Despite the heterogeneity, the main peak suggested that one or two gp120 molecules, along with one or two CD4 molecules, bind to CXCR4 tetramer, consistent with cryo-EM observations. One possible explanation for this structural variability is that CD4 binding may increase the stability of gp120, favoring the presence of two gp120 molecules in the complex.”

Methods:

“Native PAGE

The expression and purification of CXCR4, CXCL12-CXCR4, HIV-2 gp120 and CD4-ECD were same as described above except that eluted CXCR4 was subjected to size-exclusion chromatography (SEC) using a Superdex 200 Increase 10/300 GL column (GE Healthcare). Moreover, CXCR4 (calculated as monomer) was assembled with HIV-2 gp120 or gp120-CD4 at a 1:6 molar ratio to guarantee the maximum assembly efficiency. Purified CXCR4, CXCL12-CXCR4, CXCR4-gp120 and CXCR4-gp120-CD4 proteins (totally 3µg each) were loaded into native PAGE (4 to 16%, Thermo Scientific). The gel was run in low temperature (4°C) with 150 V Constant for 60 mins and then 250 V Constant for another 60 mins. The gel was fixed in the solution (40% methanol, 10% acetic acid) and heated for 45 seconds. After shaking for 15 minutes at room temperature, destain solution (8% acetic acid) was added and the gel was heated for 45 seconds. The bands were recorded by using Gel Doc XR+ imaging system (Bio-Rad) when the desired background is obtained.

Mass photometry

Mass photometry experiments were performed using a Refeyn TwoMP instrument. A pre-assembled 6-well sample cassette (Refeyn) was placed at the centre of a clean sample carrier slide (Refeyn), with each well designated for an individual measurement. To establish the focal point, 15 µL of freshly prepared SEC buffer (25 mM HEPES pH 7.5, 150 mM NaCl, 0.001% LMNG, 0.0001% CHS and 0.00033% GDN) were added to the well. The focus was determined and maintained throughout the measurement using an autofocus system based

on total internal reflection. Purified CXCR4, CXCR4-gp120^{HIV-2} and CXCR4-gp120^{HIV-2}-CD4 proteins, prepared in the same SEC buffer, were initially diluted to 200 nM and 3 μ L of the diluted sample was added to the buffer drop, achieving a final protein concentration of 33.3 nM. Once the autofocus stabilised, movies were recorded for 60 seconds. Data were collected using Refeyn AcquireMP (version 2024.1.1.0) and analysed with Refeyn DiscoverMP (version 2024.1.0.0). Contrast-to-mass calibration was carried out using a BSA protein standard (Sigma), which contained BSA monomers and dimers with molecular masses of 66.5 and 132 kDa, respectively. Statistical analysis was conducted using DiscoverMP, where Gaussian fitting was applied to distribution peaks to determine the average molecular mass of each component. Plotting was carried out using GraphPad Prism 10."

Supplementary Fig. 5, panels c-g:

c Blue native PAGE (BN-PAGE) analysis reveals the formation of CXCR4-gp120^{HIV-2} complex.

d BN-PAGE analysis of CXCR4 complexes with gp120^{HIV-2}-CD4 reveals the formation of distinct complexes.

e-g Mass photometry analyses of ligand-free or ligand-bound complexes reveal molecular masses corresponding to CXCR4 tetramer (~272 kDa, **e**), CXCR4-gp120^{HIV-2} (~365 kDa, **f**), and the ternary CXCR4-gp120^{HIV-2}-CD4 complex (~417 kDa, **g**), with buffer blank signals highlighted by asterisks.

3. In the Methods section, please provide detailed information regarding the specific HIV strains used for the gp120 protein preparation.

Thank you for pointing this out. The HIV-2 gp120 used in this study is a biological variant of HIV-2, referred to as HIV-2/vcp. This variant was derived from the HIV-2/NIH isolate and is capable

of efficiently utilizing CXCR4 independently of CD4. The sequences of gp120^{HIV-1} and gp120^{HIV-2} used for complex assembly have been included in **Supplementary Fig. 8, panels c and d**, we have also updated the methods, and the references accordingly in the revised manuscript.

Methods:

“Protein expression and purification

For recombinant expression in 293F cells, the gene of full-length human CXCR4 was cloned into pcDNA3.4 vector with a C-terminal Flag tag. Genes of human CXCL12, CD4-extracellular domain (ECD, residues 26-388), gp120^{HIV-1} (from the 92BR020 strain, residue 30-505, Uniprot: B2LT42)²¹ and gp120^{HIV-2} (from the VCP strain, residues 23-501, Uniprot: Q70145)^{50,59} were fused with interleukin-2 (IL-2) secretion signal peptides at their N terminus and His tags at their C terminus, and then cloned into pcDNA3.4 vector separately.”

Supplementary Fig. 8, panels c and d:

c

HIV2-gp120 (UniProt ref.:B2LT42)

MYRMQLLSICIALSLALVTNSKQFVTVFYGIPAWRNASIFLFCATKNRDTWGTIQCLPDNDYQEIALNVTEAFDAWNNTV
TEQAVEDVWNLFETSIPKCVKLTPLCVAMNCTRNMSTGTTDTQNIINDTSPCVRADNCTGLKEEEMVDCQFNMTGL
ERDKRKQYTEAWYSKDVICDNTSSRSKCYMNHCONTSVITESCDKHWDAMRFRYCAPPGFALLRCNDTNYSGFAPN
CSKVVAATCTRMETQSSTWFGFNGTRAENRTYIYWHGKNNRTIISLNNFYNLTMHCKRPGNKTVLPIMSGFKFHSKPV
INKKPRQAWCWFKGEWKEAMQEVKETLAKHPRYKGNRSRTENIKFKAPGRGSDPEAAYMWTNCRGEFLYCNMTWFL
NWVDNRTGQKQRNYAPCHIRQIINTWHRVGNVYLPREGELTCNSTVTSIIANIDTGDQTDITFSAEVAELYRLELGDYK
LVEITPIGFAPTSVKRYSSAHRHTRHHHHHH*

IL2-signal peptide – gp120^{HIV-2}(vcp strain, residues 23-501) – 6×His

d

HIV1-gp120 (UniProt ref.:Q70145)

MYRMQLLSICIALSLALVTNSNLWVTVYYGVPVWKEATTLFCASDAKAYKAEVHNWATHACVPTDPNPQEI VLENVTE
NFMWKNMVEQMHEDIISLWDQSLKPCVKLTPLCVTLNCLDNNSTNNNNSSGVKTGIDKGEIKNCSFNNTTTSVKDKEK
KEYALFYNLDDVQIGNDNTSYRLTSCNTSVITQACPKVTFEPIPIHYCTPAGYAILKCNKGFNGTGPCTNVSTVQCTHGI
KPVVSTQLLNGSLAEEDIVIRSENLTNNAKTIIVQLKDPVDINCTRPNNNTRKSIHIGPGRAFYATGDIIGDIRQAHCNLSRA
QWNDTLSKIVTKLREQFENKTIKQPPSSGGDPEIVFHSFNCGGEFFYCNTTQLFNSTWTNNTGTSNTTGNNDITLPCRI
KQIVNMWQEVGKAMYAPPIKGIKICSSNITGLLLTRDGGNNEMNTTEIFRPGGGDMRDNRSELYKYKVVRIEPLGIAPT
RAKRRVVQREKRHHHHHH*

IL2-signal peptide – gp120^{HIV-1}(isolate 92BR020, residues 30-505) – 6×His

c The HIV-2 gp120 construct from the VCP strain, comprising residues 23-501, was used in this study (UniProt: B2LT42).

d The HIV-1 gp120 construct from the 92BR020 strain, comprising residues 30-505, was used in this study (UniProt: Q70145).

References:

50. Endres, M. J. et al. CD4-independent infection by HIV-2 is mediated by Fusin/CXCR4. *Cell* **87**, 745-756 (1996).

59. Lin, G., Lee, B., Haggarty, B. S., Doms, R. W. & Hoxie, J. A. CD4-independent use of Rhesus CCR5 by human immunodeficiency virus Type 2 implicates an electrostatic interaction between the CCR5 N terminus and the gp120 C4 domain. *J Virol* **75**, 10766-10778 (2001).

4. In the cryo-EM sample preparation and data collection section, the authors report using Quantifoil Au 300 mesh R1.2/1.3 for the CXCR4 apo sample, which conflicts with an earlier description that mentioned holey-gold UltrAufoil grids in results section. This discrepancy should be double-checked for consistency. Additionally, the data processing section should specify the symmetry imposed during the final refinement for each dataset. For the CXCR4-gp120HIV-2 sample, the flowchart shows that particles were refined using both bin2 and bin1. This detail should be explicitly stated in the methods. Finally, the description of the local refinement for the V3 loop is unclear. Including a masked map that highlights the focused region in the cryo-EM processing flowchart would greatly improve clarity.

We appreciate the reviewer's valuable suggestions. In response, we have clarified the usage of grids throughout the text and revised the data processing section to specify the symmetries applied to each dataset, as presented in **Supplementary Fig. 2, panel b** and **Supplementary Fig. 3, panel c**. Additionally, we corrected the bin1/bin2 designation for the CXCR4-gp120^{HIV-2} dataset in the "Cryo-EM data processing" Methods section, and included the mask used for the local refinement of the CXCR4-gp120 V3 loop dataset in the cryo-EM workflow diagram shown in **Supplementary Fig. 6, panel b** of the revised manuscript.

Results:

"Overall structure of human CXCR4 tetramer

Preliminary cryo-EM analysis revealed that the CXCR4 particles exhibited severe preferred orientation on holey-carbon gold grids (Quantifoil Au 300 mesh R1.2/1.3). To address this issue, we added 0.5 mM detergent octyl maltoside fluorinated (FOM) to the solution prior to vitrification of cryo-grids."

Methods:

"Cryo-EM data processing

CXCR4 dataset: "From the best class, 734,989 particles were selected for homogeneous refinement, followed by two rounds of non-uniform refinement with a symmetry change from C1 to C4, resulting in an EM map with an average resolution of 2.87 Å.

CXCR4-CXCL12 dataset: "These particles were used for ab-initio reconstruction and heterogeneous refinement, with 135,936 particles chosen from the best class for homogeneous and two non-uniform refinement steps with a symmetry transition from C1 to C4, leading to a final reconstruction with a 3.37 Å resolution. For the CXCL12-CXCR4 structure with a 8:4 stoichiometry, a similar process was followed, and 181,863 particles were used for non-uniform refinement under C4 symmetry, producing a map resolution of 3.46 Å.

CXCR4-gp120^{HIV-2} dataset: "Each model was individually refined using homogeneous and non-uniform techniques with bin2 to enhance map density. Specifically, 41,274 particles from

class 2 were refined to 4.10 Å, 40,135 particles from class 3 were refined to 4.02 Å, 38,914 particles from class 4 were refined to 3.99 Å, and 38,098 particles from class 5 were refined to 4.10 Å. Particles from class 4 were re-extracted at bin1 and subjected to homogeneous refinement, followed by local refinement using a mask that included the CXCR4 tetramer and the V3 loop of gp120, excluding detergent micelles. This process yielded a 3.51 Å map, enabling precise modeling of the V3 loop structure.

Supplementary Fig.2, panel b:

b Image processing workflow of CXCR4 tetramer.

Supplementary Fig.3 panel c:

c Image processing workflow of CXCL12-CXCR4 complex.

Supplementary Fig.6 panel b:

b Image processing workflow of CXCR4-gp120^{HIV-2} complex.

Reviewer #2

In this manuscript, the authors presented several CXCR4 structures including homo-tetramers without ligand or G protein, CXCR4 tetramers bound to ligand (CXCL12 monomer or dimer), and CXCR4 bound to HIV-2 env gp120. A characteristic feature of this work is the CXCR4 tetramer, which is inactive (not coupled to G proteins) and able to transit to monomers with biological functions. The bindings of CXCL12 dimers and the HIV-2 gp120 further extend the manuscript that advances the research field with solid structural evidence. There are, however, a number of concerns that should be addressed by the authors.

We thank the reviewer for their valuable insights. We have addressed the concerns one by one in detail below, incorporating additional experiments, structural analyses, and citations of relevant publications reporting functional analyses. We have also included a discussion addressing the limitations where we were unable to fully resolve certain questions related to the functional study.

Specific comments:

1, All CXCR4 structures obtained are tetramers, whereas in other published reports monomers (Gi protein-coupled), dimers and trimers are found. Sample preparation methods should be compared and any differences identified. Did the inclusion of a C terminal FLAG tag play a role in tetramer formation?

We thank the reviewer for the helpful suggestion. We carefully analyzed the differences in CXCR4 sample preparation reported in previous studies (**Response Table 1**, see below) and re-examined our CXCR4 apo sample using native PAGE. We also reanalyzed the cryo-EM datasets to assess CXCR4 apo and CXCL12-CXCR4 oligomerization. Based on these analyses, we discussed potential reasons for the observed differences in CXCR4 oligomerization compared to previously published studies.

Comparison of CXCR4 Conformational States:

Monomeric State: The active, Gi protein-coupled state of CXCR4 consistently adopts a monomeric conformation, as shown in recent cryo-EM studies (PDB: 8K3Z, PMID: 39093700; PDB: 8U4O, PMID: 39313635). Structural alignment suggests that the active state of CXCR4 cannot exist as a higher-order oligomer (such as a tetramer) due to steric clashes between the C-terminal helix (residues G306-T318; see **Extra figure, panel a**) and adjacent protomers in the tetrameric assembly. CXCR4 alone also exists in a monomeric state when expressed in insect cells and purified using a Superdex 200 Increase column (PMID: 39313635). In our experiments using the same column, a minor peak at 13 mL suggested a monomeric population, which was further validated by Native-PAGE and de-glycosylated analysis (**Supplementary Fig. 1**). Moreover, 2D classification of our CXCR4 cryo-EM dataset revealed that approximately 3% of particles (8,035 out of 182,049) corresponded to the monomeric state, supporting its presence in the sample (**Supplementary Fig. 2, panel d**).

Dimeric State: Dimeric assemblies of CXCR4 have only been observed in crystal structures (PDB: 3ODU, 3OE0), where the receptor was expressed in insect cells and purified with DDM detergent (PMID: 20929726).

Trimeric State: Trimeric CXCR4 structures were detected when the receptor was mixed with REGN7663 Fab prior to cryo-EM grid preparation (PMID: 39313635). This is currently the only reported condition yielding a trimer, suggesting that either the Fab induces trimer formation or that the trimer exists in very low abundance under native conditions.

Tetrameric State: Tetrameric CXCR4 conformations were observed both in the presence of REGN7663 Fab (PMID: 39313635) and in ligand-free receptor samples expressed in insect cells (PMID: 39891908). These findings are consistent with our own data, where tetramers were detected in CXCR4 samples expressed in mammalian cells.

Notably, single-molecule fluorescence studies have shown that CXCR4 exists in a dynamic equilibrium among monomeric, dimeric, and higher-order oligomeric states in live cells (PMID: 28118546; PMID: 29625032). In our apo samples, we detected monomeric and tetrameric forms but no clear evidence of dimeric or trimeric states. This may reflect differences in the lipid environment associated with the expression system (e.g., insect cells vs mammalian cells) and its influence on CXCR4 oligomerization (PMID: 39313635; PMID: 39891908).

Our CXCR4 construct includes a C-terminal Flag tag, a commonly used strategy in GPCR expression that is unlikely to influence tetramer formation. Indeed, previous reports have observed monomeric, trimeric, and tetrameric states using a similar CXCR4-monomeric eGFP-Flag-C construct (PMID: 39313635). Likewise, an N-terminal Flag tag was used in a study that observed CXCR4 tetramers (PMID: 39891908), further supporting the biological relevance of these states. Finally, the use of fluorinated octyl maltoside (FOM) to mitigate preferred orientation issues in cryo-EM does not appear to artificially induce CXCR4 tetramerization or head-to-head CXCR4-CXCL12 tetramer formation. The head-to-head CXCR4-CXCL12 tetramer assemblies were observed in 2D classes even in samples prepared without FOM (see **Extra figure panel b**) and were independently confirmed by Native-PAGE performed on FOM-free samples (**Supplementary Fig. 3, panel e**). We have updated these analyses in the Results section, **Supplementary Fig. 1, Supplementary Fig. 2, panel d** and **Supplementary Fig. 3, panel e** of the revised manuscript.

Results:

“Overall structure of human CXCR4 tetramer

The elution volume of gel filtration suggested that CXCR4 primarily exists as an oligomer. Additionally, we observed a minor peak following the main peak. SDS-PAGE analysis confirmed that both peaks correspond to CXCR4 protein (Supplementary Figs. 1a, b). Native page showed that has an apparent molecular weight of approximately 120 kDa, while the main peak is around 480 kDa- roughly four times larger. Based on these observations and consistent with a recent publication²⁵, we assigned these two peaks to monomeric and tetrameric forms of CXCR4, respectively (Supplementary Figs. 1d, e). The rising molecular weights observed on the native gel are likely influenced by detergent micelles surrounding the CXCR4 proteins.

On the SDS page, we observed diffuse CXCR4 bands, which shifted to a low molecular weighted upon treatment with PNGase F-an enzyme for removing almost all N-linked oligosaccharides from glycoproteins, suggesting that CXCR4 is highly glycosylated (Supplementary Fig. 1f).

2D classification identified a minor population (~3%) of monomeric CXCR4 (Supplementary Fig. 2d). However, the limited particle number and size constraints of cryo-EM prevented us from reconstructing a high-resolution structure.

Two distinct assembly mechanisms of CXCL12-CXCR4 complex formation

We obtained two distinct structures of the CXCL12-CXCR4 complex (**Supplementary Figs. 3b-d**), namely with 8:4 stoichiometry at 3.4 Å (**Fig. 2a**) and 8:8 stoichiometry at 3.3 Å (**Fig. 2b**). Both states were confirmed by native PAGE analysis (**Supplementary Fig. 3e**)."

Methods:

“Native PAGE

The expression and purification of CXCR4, CXCL12-CXCR4, HIV-2 gp120 and CD4-ECD were same as described above except that eluted CXCR4 was subjected to size-exclusion chromatography (SEC) using a Superdex 200 Increase 10/300 GL column (GE Healthcare). Moreover, CXCR4 (calculated as monomer) was assembled with HIV-2 gp120 or gp120-CD4 at a 1:6 molar ratio to guarantee the maximum assembly efficiency. Purified CXCR4, CXCL12-CXCR4, CXCR4-gp120 and CXCR4-gp120-CD4 proteins (totally 3µg each) were loaded into native PAGE (4 to 16%, Thermo Scientific). The gel was run in low temperature (4°C) with 150 V Constant for 60 mins and then 250 V Constant for another 60 mins. The gel was fixed in the solution (40% methanol, 10% acetic acid) and heated for 45 seconds. After shaking for 15 minutes at room temperature, destain solution (8% acetic acid) was added and the gel was heated for 45 seconds. The bands were recorded by using Gel Doc XR+ imaging system (Bio-Rad) when the desired background is obtained.

Deglycosylation assay

PNGase F (P0704S, NEB) was used for removing N-linked oligosaccharides from CXCR4. Briefly, 0.42 µg CXCR4 proteins (tetramer or monomer) were mixed with 1× GlycoBuffer 2 and 0.5 µl PNGase F to a total reaction volume of 20 µL. Reaction without PNGase F served as negative controls. After incubation at 37°C for 1.5 hour, the separation of reaction products was analyzed by western blot experiments using HRP-conjugated antibodies against Flag-tag (1:2,000-fold dilution (v/v)). The bound antibodies were visualized by using an enhanced chemiluminescence reagent and quantified by densitometry using a ChemiDoc MP imaging system (Bio-Rad)."

Supplementary Fig.1 panels a, b, d and f:

a Size exclusion chromatography (SEC) of purified CXCR4 expressed in mammalian cells showing distinct peaks corresponding to aggregates, tetramers, monomers, and free Flag peptides.

b,c SDS-PAGE analysis confirms the monomeric (b) and tetrameric (c) CXCR4 bands related to a.

d,e Blue native PAGE (BN-PAGE) analysis showing distinct bands for CXCR4 monomer (d) and tetramer (e).

f Western blot analysis with and without PNGase F treatment confirms glycosylation of CXCR4 in both monomeric and tetrameric forms.

Supplementary Fig.2 panel d:

d Class averages showing a minor population of monomeric CXCR4 particles. From 182,049 total particles in this subset, 8,035 particles (~3%) were classified as monomeric (highlighted in red boxes).

Supplementary Fig.3 panel e:

e BN-PAGE analysis of CXCL12-CXCR4 complexes reveals the formation of both 8:4 and 8:8 stoichiometries when assembled at a 1:6 molar ratio (monomer basis).

Extra figure for reviewer's consideration:

a The CXCR4 tetramer structure was aligned with monomeric CXCL12-CXCR4-Gi-scFv16 (8K3Z) and CXCL12-CXCR4-Gi (8U4O), showing high similarity of the CXCR4 monomers with an RMSD of 0.569 Å. The zoomed-in box and arrow show that the C-terminal helix (residues G306 to T318) clashes with a neighboring subunit in the CXCR4 tetramer.

b 2D classification of the CXCL12-CXCR4 complex in the absence of FOM. Selected 2D class averages are shown. Particles highlighted with red boxes correspond to the dimeric tetramer form of the CXCL12-CXCR4 complex (8:8 stoichiometry).

Response Table 1: “**Comparison of CXCR4 Conformational States**”

Oligomeric states	Structure /complex	Expression strategy	Expression system	Detergent for SEC	Oligomeric analysis
Monomer	CXCL12-CXCR4-Gi-scFv16	Co-expression of N-Flag-BRIL-CXCR4, CXCL12-8xHis-C, G α i, G β 1, and G γ 2; followed by incubation with CXCL12 and scFv16 and SEC purification	Insect cells	0.00075% LMNG, 0.0002% CHS	Cryo-EM (PMID:39093700) 8K3Z
	CXCR4_{EM}-Gi-CXCL12	Co-expression of CXCR4-monomeric eGFP-Flag-C, His-tagged G α i, and His-tagged G β 1-G γ 2; incubated with CXCL12 prior to grid preparation	Insect cells	0.01% LMNG, 0.001% CHS	Cryo-EM (PMID:39313635) 8U4O
	HF51116/AMD070/AMD3100 - CXCR4^{κOR}-Nb6	Purified N-Flag-BRIL-CXCR4 (L125W, T240P; ICL3 replaced by κ OR segment)-10xHis-C with antagonist; incubated with Nb6-8xHis-C and purified by SEC	Insect cells	0.01% LMNG, 0.002% CHS	Cryo-EM (PMID:40063796) 8ZPL 8ZPM 8ZPN
	CXCR4	CXCR4- monomeric eGFP-Flag-C	Insect cells	0.01% LMNG, 0.001% CHS	SEC and Native-PAGE (PMID:39313635)
	CXCR4	CXCR4-Flag-C	Expi293 cells	0.01% LMNG, 0.001% CHS, 0.00033% GDN,0.5mM FOM	SEC,Native-PAGE and Mass photometry (this study)
Dimer	CXCR4-ITIt/CVX15	Purified N-Flag-3C-CXCR4-10xHis-C in the presence of antagonist	Insect cells	0.05%DDM, 0.01% CHS	Crystal structure (PMID:20929726) 3ODU 3OE0
Trimer	CXCR4_{EM}-REGN7663 Fab	CXCR4-monomeric eGFP-Flag-C mixed with REGN7663 Fab prior to grid preparation	Insect cells	0.01% LMNG, 0.001% CHS	Cryo-EM (PMID:39313635) 8U4S 8U4T
	CXCR4_{EM}-REGN7663 Fab				
Tetramer	CXCR4	N-Flag-CXCR4	Insect cells	0.00075% LMNG, 0.0002% CHS	Cryo-EM (PMID:39891908) 8YU7
	CXCR4	CXCR4-Flag-C	Expi293 cells	0.01% LMNG, 0.001% CHS, 0.00033% GDN, 0.5 mM FOM	Cryo-EM (this study)

2, The authors found that the N terminus of CXCR4 was flexible, and a cryo-EM structure could not be obtained. This is in contrast to general believe that the N terminus of the chemokine receptor wraps around the core structure of the chemokine. Could the observed phenomenon a feature specific to the tetrameric CXCR4?

Thank you for your insightful comment. As you noted, the N-terminus of chemokine receptors can directly engage with the chemokine core, as demonstrated in the CCR5-MIP-1 α complex (PDB 7F1T). However, in the case of CXCR4, the N-terminus is consistently unresolved or only partially resolved across all reported structures, including monomeric, dimeric, trimeric, and tetrameric forms.

In the monomeric CXCR4-CXCL12-Gi complex structures (PDBs 8U4O and 8K3Z), the resolved portion of the N-terminus begins at M24, which is consistent with our own data. In our structures, the CXCR4 apo form and the CXCR4-CXCL12 complex both begin at K25, while the CXCR4-gp120^{HIV-2} and the CXCR4-gp120^{HIV-2}-CD4 complexes begin at E26. A similar trend is observed in other CXCR4 structures. For example, the CXCR4-vMIP-II complex (PDB 4RWS) starts at S23. Structures bound to small molecule antagonists, such as HF51116, AMD070, and AMD3100 (PDBs 8ZPL, 8ZPM, and 8ZPN), start at A34. Other complexes, such as those with IT1t (PDB 3ODU) and CVX15 (PDB 3OE0), begin at P27 and K25, respectively. The recently reported CXCR4 tetramer structure (PDB 8YU7) also begins at E26.

Together, these data suggest that the N-terminus of CXCR4 is often unresolved in both ligand-bound and apo states. This includes complexes with chemokines (monomeric and tetrameric), viral peptides such as vMIP-II, small molecule antagonists (monomeric and dimeric), and viral glycoproteins such as HIV-2 gp120 in tetrameric form. However, we cannot conclude that the N-terminus of CXCR4 is inherently flexible, as the "two site" model of chemokine binding, supported by NMR structures of the CXCR4 N-terminus in complex with CXCL12 (PDBs 2N55 and 2K04), suggests that it may adopt a defined conformation under certain conditions. We have removed the phrase "likely due to inherent flexibility" and revised the main text to reflect the updated discussion in the revised manuscript.

Results:

"Two distinct assembly mechanisms of CXCL12-CXCR4 complex formation"

The N-terminus of CXCR4 acts as the chemokine recognition site 1 (CRS1), engaging with the globular core of CXCL12, while the seven transmembrane helices of CXCR4 form chemokine recognition site 2 (CRS2), which accommodates the insertion of the N-terminus of CXCL12 into a pocket ³⁸⁻⁴⁰. In our structure, similar to the CXCR4 tetramer alone, the N-terminal residues (1-25) of CXCR4 in the CXCL12-CXCR4 complex are not visible, which is consistent with all reported CXCR4 ligand-bound and apo states, including complexes with chemokines (PDBs: 8U4O and 8K3Z)^{25,34}, viral peptides (e.g. vMIP-II, PDB: 4RWS)⁴¹, small-molecule antagonists (PDBs: 8ZPL, 8ZPM, 8ZPN)⁴².

3, No functional assays were included in this manuscript. Did the R12A mutation affect function of CXCL12 in activation of CXCR4?

Thanks for your suggestion. We have not directly assessed whether the R12A mutation in CXCL12 affects CXCR4 activation. However, several publications suggest that substitution of

R12 with alanine severely compromises CXCR4-mediated mediated intracellular calcium signaling and chemotactic responses. We have included a brief description and the relevant references in the revised manuscript.

Results:

“Two distinct assembly mechanisms of CXCL12-CXCR4 complex

In contrast, a mutation at R12, or simultaneous mutations in all interaction residues, completely abolish interactions between CXCL12 and CXCR4 (**Fig. 2d**). The R12_{CXCL12} mutation was also reported to cause a dramatic loss of potency in CXCR4-mediated intracellular calcium signaling and chemotactic responses⁴⁴⁻⁴⁶.

References:

44. Ziarek, J. J. et al. Structural basis for chemokine recognition by a G protein-coupled receptor and implications for receptor activation. *Sci Signal* **10**, 471(2017).
45. Murphy, J. W. et al. Structural and functional basis of CXCL12 (stromal cell-derived factor-1 alpha) binding to heparin. *J Biol Chem* **282**, 10018-10027 (2007).
46. Ohnishi, Y. et al. Crystal structure of recombinant native SDF-1 α with additional mutagenesis studies:: An attempt at a more comprehensive interpretation of accumulated structure-activity relationship data. *J Interf Cytok Res* **20**, 691-700 (2000).

4, The binding of CXCL12, whether in monomer or dimer, induces transmembrane signaling through CXCR4. Signaling assays are extensively used in structural studies to test and confirm agonist binding, often with mutations of key binding residues. It is suggested that authors include CXCR4-based signaling assays with receptor mutants to confirm their models. This is especially important in consideration that CXCL12 dimer activates CXCR4 in a different way from CXCL12 monomer, as reported in Ref# 29. The binding to vs. activation of CXCR4 by both CXCL12 monomer and dimer should be examined using approaches reported in that reference, in order to find any structural difference underlying the signaling difference.

Thank you for your thoughtful suggestion to validate our model using CXCR4 mutants in CXCL12-mediated signaling. Indeed, mutations at key residues identified in our CXCR4-CXCL12 structures, such as E268 and E288, which contribute to salt bridge formation, and W94, Y116, and F189, which are involved in cation- π or π - π interactions with CXCL12, which have been reported to significantly impair calcium flux signaling (PMIDs: 27543332, 10825158, and 11551942).

We agree that the differential activation of CXCR4 by CXCL12 dimers versus monomers is an important aspect of this signaling pathway. However, there are a few reasons why we believe it may not be appropriate to include these experiments in the current manuscript. First, the primary aim of our study is not to dissect the functional role of specific binding residues in CXCR4 activation. Rather, our focus is to demonstrate that CXCL12 occupies the same binding pocket as HIV gp120 and that they share a conserved binding module. Second, we currently lack an optimized system for conducting functional assays, and establishing one would require substantial time and effort. Given the importance of our structural findings, we prefer to publish them without delay. Nonetheless, we value your suggestion and plan to pursue these functional studies in future work, if necessary, through collaboration.

5, Is there experimental proof that CXCR4 tetramers exist in intact cells?

To the best of our knowledge, there is no direct evidence demonstrating the existence of CXCR4 tetramers in intact living cells. However, several studies have reported that CXCR4 can exist as monomers, dimers, oligomers, or even basal nanoclusters under physiological conditions. For example, one group combined bioluminescence resonance energy transfer (BRET) with bimolecular fluorescence complementation (BiFC) to determine receptor stoichiometry in living cells, providing the first evidence for CXCR4 multimerization (PMID: 19018754). Another study using single-molecule imaging reported that CXCR4 predominantly exists as monomers in normal mammalian cells but forms dimers and higher-order oligomers at elevated expression levels, such as in cancer cells (PMID: 28118546). Additionally, single-particle tracking and super-resolution microscopy revealed that CXCR4 can form basal nanoclusters in resting T cells (PMID: 29625032). While these findings support the presence of oligomeric or higher-order organizations of CXCR4, none directly confirm the presence of tetramers in intact living cells. We believe that the dynamic oligomeric states of CXCR4 may underlie distinct signaling mechanisms and contribute to diverse cellular outcomes.

6, How did CXCL12a and CXCL12b interact with each other? Providing additional details of the molecular interaction could be helpful.

Thank you for pointing this out. We compared our CXCL12 dimer structure with those in PDB entries 1QG7, 1A15, and 2K01, which represent the mammalian crystal structure, a synthetic structure, and an NMR structure, respectively. The CXCL12 dimer in our structure closely resembles both the crystal and NMR structures, except for two regions: the N-terminal segment, which is critical for receptor interaction, and the C-terminal helix (**Supplementary Fig. 4, panel d**). We also identified the residues at the CXCL12 dimer interface, which is primarily stabilized by main-chain interactions between the β 1 strands of the two CXCL12 monomers. In addition, we present the local density at the dimer interface, as well as around the β 2 and β 3 strands (**Supplementary Fig. 4, panels b and c**). These details have been included in the main text and in **Supplementary Fig. 4, panels b-d** of the revised manuscript.

Results:

“Two distinct assembly mechanisms of CXCL12-CXCR4 complex formation

The CXCL12 dimer interface in our structures is primarily mediated by main-chain interactions between the β 1 strands of the two monomers (**Supplementary Figs. 4b, c**), closely resembles both the crystal and NMR structures⁴⁸⁻⁵⁰ (PDB IDs: 1QG7, 1A15, and 2K01, corresponding to a mammalian crystal form, a synthetic variant, and an NMR model, respectively). The CXCL12 dimer in our structure closely resembles, except for two regions: the N-terminal segment, which is critical for receptor interaction, and the C-terminal helix (**Supplementary Fig. 4d**).”

Supplementary Fig. 4, panels b-d:

b CXCL12 dimer interface, with labeling of interfacial residues.

c Local density of the CXCL12 dimer interface, emphasizing beta1. The local density of beta2 and beta3 is also presented.

d Comparison of the CXCL12 dimer in our structure (in color yellow and orange) with those in PDB entries 1QG7 (in dark green), 1A15 (in light green), and 2K01 (in silver), which represent the mammalian crystal structure, a synthetic structure, and an NMR structure, respectively, showing notable differences at the N-terminal loop and C-terminal helix.

Minor comments:

1, The authors are suggested to compare their tetramer structure with recently reported CXCR4 tetramer from other groups (Saotome et al. 2025, PMID 39313635; Liu et al. 2025, PMID 39891908) and comment on similarities and differences if any.

We superimposed the reported CXCR4 tetrameric structures (PDB: 8U4T, 8YU7) onto our CXCR4 tetramer structure and summarized the similarities and differences in the main text and **Supplementary Fig. 2, panel e** of the revised manuscript.

Results:

“Overall structure of human CXCR4 tetramer

The structure of CXCR4 tetramer is nearly identical to previously reported tetrameric CXCR4 structures (PDB: 8YU7 and 8U4T)^{25,33}, with RMSD values of 0.840 Å and 0.871 Å,

respectively. Only minor differences are observed in the N-terminus and the ECL2 region of CXCR4, which are responsible for the interactions with the Fab (**Supplementary Fig. 2e**).”

Supplementary Fig. 2 panel e:

e Structural alignment of the CXCR4 tetramer (magenta) with the CXCR4-REGN7663 complex (PDB: 8U4T, light gray) and CXCR4 tetramer (PDB: 8YU7, dark gray), showing high overall similarity (RMSD = 0.871 Å and 0.840 Å, respectively). The central panel displays the aligned structures, with Fab fragments (REGN7663) shown in light blue and purple. Zoomed-in views highlight minor structural differences at the N-terminus (left panel) and ECL2 loop (right panel), which are involved in Fab binding.

2, While theoretically 4 HIV-2 gp120 env may be bound to the CXCR4 tetramer, the authors observed only one gp120 binding. Could this be the result of steric clash as the gp120 protein is quite large?

Thank you for pointing this out. The CXCR4-gp120^{HIV-2} complex was initially prepared at a 1:2 molar ratio of CXCR4 (calculated as a monomer) to gp120 and incubated for 2 hours on ice. The resolved structure shows that the CXCR4 tetramer binds only one gp120 molecule. Although the 2D class averages (**Supplementary Fig. 6, panel b**) suggest the presence of two gp120 molecules bound on opposite protomers of a single CXCR4 tetramer, this is not consistently observed in the final reconstruction.

To rule out the possibility that an insufficient gp120 ratio was limiting binding (e.g., preventing binding of three or four gp120 molecules per tetramer), we assembled the CXCR4-gp120^{HIV-2} and CXCR4-gp120^{HIV-2}-CD4 complexes at a 1:6 molar ratio (i.e., an excess of gp120 or gp120^{HIV-2}-CD4, corresponding to six gp120^{HIV-2} or gp120^{HIV-2}-CD4 complexes per CXCR4 tetramer) at 4 °C overnight. The molecular weights of the resulting complexes were analyzed using native PAGE and mass photometry. Mass photometry revealed that the CXCR4-gp120^{HIV-2} complex exhibited a MW increase of approximately 93 kDa compared to the CXCR4 tetramer alone (272 kDa), suggesting the binding of one or two gp120 molecules (each ~58.2 kDa) to the CXCR4 tetramer (**Supplementary Fig. 5, panels e and f**). For the CXCR4-gp120^{HIV-2}-CD4 complex, the MW increase was ~145 kDa, while the gp120^{HIV-2}-CD4 subcomplex has a theoretical MW of ~102 kDa. This suggests the presence of one or two gp120^{HIV-2}-CD4 subcomplexes bound to the CXCR4 tetramer, consistent with our cryo-EM data, which show either one or two gp120 molecules bound to the CXCR4-gp120^{HIV-2} complex, and two gp120^{HIV-2}-CD4 molecules bound within the CXCR4-gp120^{HIV-2}-CD4 complex. However, structural modeling suggests that the CXCR4 tetramer can accommodate four gp120^{HIV-2} proteins without any apparent steric clashes, indicating that the complex with a single gp120^{HIV-2} bound may be more stable than other configurations, at least

when only gp120^{HIV-2} is present. Additionally, native PAGE showed corresponding band shifts for these complexes (**Supplementary Fig. 5, panels c and d**). These results have been updated in the main text, Methods section, and **Supplementary Fig. 5, panels c-g** of the revised manuscript.

Results:

“Reconstitution of CXCR4-gp120^{HIV-2}-CD4 complex

Native page gel results showed a broad distribution of the CXCR4-gp120^{HIV-2} and CXCR4-gp120^{HIV-2}-CD4 complex (**Supplementary Figs. 5c, d**), making it is difficult to determine the exact stoichiometry of gp120 and CD4 binding to CXCR4 tetramer. To further clarify this, we employed mass photometry to assess their molecular weights (**Supplementary Figs. 5e-g**). Despite the heterogeneity, the main peak suggested that one or two gp120 molecules, along with one or two CD4 molecules, bind to CXCR4 tetramer, consistent with cryo-EM observations. One possible explanation for this structural variability is that CD4 binding may increases the stability of gp120, favoring the presence of two gp120 molecules in the complex.”

Methods:

“Native PAGE

The expression and purification of CXCR4, CXCL12-CXCR4, HIV-2 gp120 and CD4-ECD were same as described above except that eluted CXCR4 was subjected to size-exclusion chromatography (SEC) using a Superdex 200 Increase 10/300 GL column (GE Healthcare). Moreover, CXCR4 (calculated as monomer) was assembled with HIV-2 gp120 or gp120-CD4 at a 1:6 molar ratio to guarantee the maximum assembly efficiency. Purified CXCR4, CXCL12-CXCR4, CXCR4-gp120 and CXCR4-gp120-CD4 proteins (totally 3µg each) were loaded into native PAGE (4 to 16%, Thermo Scientific). The gel was run in low temperature (4°C) with 150 V Constant for 60 mins and then 250 V Constant for another 60 mins. The gel was fixed in the solution (40% methanol, 10% acetic acid) and heated for 45 seconds. After shaking for 15 minutes at room temperature, destain solution (8% acetic acid) was added and the gel was heated for 45 seconds. The bands were recorded by using Gel Doc XR+ imaging system (Bio-Rad) when the desired background is obtained.

Mass photometry

Mass photometry experiments were performed using a Refeyn TwoMP instrument. A pre-assembled 6-well sample cassette (Refeyn) was placed at the centre of a clean sample carrier slide (Refeyn), with each well designated for an individual measurement. To establish the focal point, 15 µL of freshly prepared SEC buffer (25 mM HEPES pH 7.5, 150 mM NaCl, 0.001% LMNG, 0.0001% CHS and 0.00033% GDN) were added to the well. The focus was determined and maintained throughout the measurement using an autofocus system based on total internal reflection. Purified CXCR4, CXCR4-gp120^{HIV-2} and CXCR4-gp120^{HIV-2}-CD4 proteins, prepared in the same SEC buffer, were initially diluted to 200 nM and 3 µL of the

diluted sample was added to the buffer drop, achieving a final protein concentration of 33.3 nM. Once the autofocus stabilised, movies were recorded for 60 seconds. Data were collected using Refeyn AcquireMP (version 2024.1.1.0) and analysed with Refeyn DiscoverMP (version 2024.1.0.0). Contrast-to-mass calibration was carried out using a BSA protein standard (Sigma), which contained BSA monomers and dimers with molecular masses of 66.5 and 132 kDa, respectively. Statistical analysis was conducted using DiscoverMP, where Gaussian fitting was applied to distribution peaks to determine the average molecular mass of each component. Plotting was carried out using GraphPad Prism 10.”

Supplementary Fig.5 panels c-g:

c Blue native PAGE (BN-PAGE) analysis reveals the formation of CXCR4-gp120^{HIV-2} complex.

d BN-PAGE analysis of CXCR4 complexes with gp120^{HIV-2}-CD4 reveals the formation of distinct complexes.

e-g Mass photometry analyses of ligand-free or ligand-bound complexes reveal molecular masses corresponding to CXCR4 tetramer (~272 kDa, e), CXCR4-gp120^{HIV-2} (~365 kDa, f), and the ternary CXCR4-gp120^{HIV-2}-CD4 complex (~417 kDa, g), with buffer blank signals highlighted by asterisks.

3, The “sandwich” model in which CXCL12 dimers bind CXCR4 on both sides was interesting and could potentially serve a role for juxtacrine signaling, but is there any experimental evidence for its presence?

To our knowledge, there is no direct evidence for its presence. However, a recent study (PMID: 29625032) reported that CXCR4 can form basal nanoclusters in resting T cells, which may suggest the potential for CXCL12 dimers to engage CXCR4 tetramer on both sides.

4, Page 6, second paragraph: “This shift alters the configuration of CXCL12, particularly the N-terminal loop”: should be “conformation” instead of “configuration”.

Thanks. We have incorporated this change in the revised manuscript.

5, Fig 4d: HIV-2’s slower progression and lower transmissibility compared to HIV-1 hint at Env differences. Its gp120 may have distinct binding dynamics with CXCR4, potentially influencing entry dynamics. Authors may want to discuss on this to elaborate their modeled findings, and this discussion would provide more comprehensive implications of their study for understanding the distinctive properties of HIV-2 Env interacting with CXCR4.

Thank you for your insightful comment. Our model illustrates that the HIV-1 Env trimer is capable of binding only one CXCR4 tetramer, whereas the HIV-2 Env trimer can, in principle, engage up to three CXCR4 tetramers. This difference may underlie the variations in disease progression and transmissibility observed between HIV-1 and HIV-2, as discussed in the manuscript. However, how this model translates to behavior in living cells remains unclear. It is possible that both HIV-1 and HIV-2 ultimately bind only one CXCR4 tetramer in vivo, with differences in infection efficiency arising primarily from CD4 binding affinity (PMID: 2190604). At this stage, we cannot say for certain.

6, Given the relevance of CXCR4 antagonists for HIV therapeutics, Extended Data Fig. 9 should be moved to the main figure part.

Thank you for pointing this out. We have relocated Extended Data Figure 9 to **Figure 6** in the revised manuscript.

Reviewer #3

CXCR4 is a chemokine receptor that interacts with its ligand CXCL12 and plays an important role in cell migration and hematopoiesis, and it also serves as one of the HIV co-receptors. High-resolution structural information would be critical for our understanding of the molecular mechanisms of how CXCR4 mediates both cellular regulation and HIV inhibition. In this manuscript, Zhang et al. reported cryo-EM structures of CXCR4 alone, CXCR4-CXCL12 complex, CXCR4-gp120 complex and CXCR4-gp120-4D CD4 complex. In all cases, CXCR4 unexpectedly formed a tetramer, different from what have been reported previously. CXCL12 bound to CXCR4 in the same way as other chemokines bind to their receptor. The binding mode of HIV-2 gp120 to CXCR4 is also very similar to that of how HIV-1 gp120 binds to CCR5. The authors suggest that their “study provides fundamental structure-based mechanistic insights into how HIV-2 gp120 binds its co-receptor CXCR4 and explains how CXCL12 blocks HIV entry, offering guidance for the development of vaccines and therapeutic agents targeting HIV infection”. Structural studies of CXCR4 and its complexes with various ligands are highly significant and should be of great interest to readers in both the GPCR and HIV fields. There are some issues with the current version of manuscript, however.

We appreciate the reviewer’s valuable feedback and suggestions regarding the manuscript’s writing. We have carefully reviewed the text, implemented the necessary revisions, and addressed each comment in detail.

Major points:

1. While the density for CXCR4 is quite good in most of the structures except for the CXCR4-gp120-CD4 complex, the density for gp120 is rather poor (at least based on all the figures presented), even for the V3 loop that inserts into the chemokine-binding pocket (CRS2) and is supposed to be rigid. It is unclear whether their model for gp120 binding to CXCR4 would be more reliable than an AlphaFold-predicted model. It would be a good idea to tone down the discussion about the interactions at the atomic level.

We appreciate the reviewer's suggestion. In response, for CXCR4-gp120^{HIV-2} map, we added the cryo-EM density at the interface between CXCR4 CRS2 and the V3 loop of gp120^{HIV-2} in **Supplementary Fig. 6, panel d** (related to Figure 3, panel d), along with overview density for gp120 in **Supplementary Fig. 6, panel e**, which demonstrates well-resolved side chains in both CXCR4 CRS2 and the V3 loop of gp120^{HIV-2}. For CXCR4-gp120^{HIV-2}-CD4 map, we included the local density maps of the V3 loop of gp120^{HIV-2} in **Supplementary Fig. 9, panel d**, which demonstrate the reasonable side chain fitting at the CXCR4-gp120^{HIV-2} interface. Key residues in the V3 loop (e.g., K310 and H312) involved in CXCR4 recognition were validated by pull-down assays in this study. Furthermore, mutations in residues mediating hydrogen bonds or salt bridges within the CXCR4 CRS2 pocket (e.g., D262, E268, E277, and E288) led to a 40%-60% reduction in HIV-1 LAI/NDK infection compared to wild-type CXCR4 (PMID: 10825158), further supporting our structural models of gp120 in complex with its receptors.

To compare our cryo-EM structures with the AlphaFold3-predicted models, we input four full-length CXCR4 copies. The five predicted models show limited conservation of the tetrameric assembly (ipTM: 0.22, pTM: 0.36), particularly in the N-terminal domain (NTD; residues 1-27) and the C-terminal domain (CTD; residues 306-C), as shown in **Response Fig. 1, panel a**. These regions lack density in our cryo-EM structures and in previously published CXCR4 tetrameric structures (PDB: 8U4T and 8YU7), as well as in other oligomeric CXCR4 structures (PDB: 4RWS and 3OE0), supporting the inherent flexibility of the NTD and CTD of the CXCR4 protein. An exception is observed in antagonist-bound or Gi-protein-coupled CXCR4 structures, where interactions at the C-terminus may stabilize the C-terminal helix (residues 306-315), as seen in structures 8U4O, 8K3Z, 8ZPL, 8ZPM, and 8ZPN. When comparing the predicted CXCR4 tetramer models with our cryo-EM structure, the tetrameric interface is markedly different. However, each individual monomer aligns well with our cryo-EM model, with an RMSD of 0.769 Å for the CXCR4 monomer (**Response Fig. 1, panels a and b**).

Since the tetrameric CXCR4 assemblies predicted by AF3 are less reliable, we next modeled the monomeric CXCR4-gp120^{HIV-2} complex by inputting one full-length human CXCR4 molecule and one full-length gp120 from the HIV-2/VCP variant (residues 23-501, excluding the signal peptide 1-22). We also predicted the monomeric CXCR4-gp120^{HIV-2}-CD4 complex by including an additional human CD4 molecule comprising only the D1 and D2 domains (residues 26-201, excluding its signal sequence 1-25).

The five predicted models of both the CXCR4-gp120^{HIV-2} and CXCR4-gp120^{HIV-2}-CD4 complexes exhibited low interfacial predicted TM-scores (ipTM: 0.15 and 0.50) and predicted TM-scores (pTM: 0.49 and 0.54), indicating poor model consistency. Across the models, the N-terminal loop and the C-terminal helix of CXCR4 remained flexible. Furthermore, the V3 loop of gp120, which contains the GFKF motif and was identified by cryo-EM to insert into the CXCR4 CRS2 pocket, was not positioned correctly in any predicted models. Instead, the gp120^{HIV-2} molecule was variably rotated relative to CXCR4, even when the CXCR4 monomers aligned well with our cryo-EM structure-highlighting the dynamic nature of gp120 in complex with CXCR4 (**Response Fig. 1, panels c-f**). When superimposing gp120^{HIV-2}-CD4 models, the structures were highly similar,

with RMSD values of 0.475 Å, 0.447 Å, 0.471 Å, and 0.426 Å compared to model 1, indicating a conserved gp120-CD4 interface (**Response Fig. 1, panel g**). This conservation is also reflected in our lower-resolution cryo-EM map, which reveals a similar interface (RMSD: 0.819 Å vs. model 1), except for the V3 loop (**Response Fig. 1, panel h**).

In conclusion, while AF3 accurately predicts the CXCR4 monomer (including its dynamic NTD loop and CTD helix) and reproduces the general gp120^{HIV-2}-CD4 architecture consistent with our and previously reported structures, it fails to predict the tetrameric assembly. Our cryo-EM data, along with the recently reported CXCR4 tetramer (PDB: 8YU7), confirm the existence of this tetrameric interface. Most importantly, AF3 fails to model the V3 loop containing the GFKF motif—a sequence similar to the GPCR-interacting motif in HIV-1 gp120 (CCR5-gp120^{HIV-1}-CD4 complex; PDB: 6MEO, Supplementary Fig. 8, panel b), which inserts into the CXCR4 CRS2 pocket. Instead, AF3 predicts either non-conserved β-hairpins or no insertion at all, thereby missing this critical feature that competes with GPCR activation, including ligand binding (e.g., CXCL12) or antagonist and peptide binding (e.g., IT1t, vMIP-II). We have updated the local density map fitting in **Supplementary Fig. 6, panels d and e**, and **Supplementary Fig. 9, panel d** in the revised manuscript.

Supplementary Fig. 6, panels d and e:

d Representative local density maps of CXCR4-gp120HIV-2 interface.

e Overview density map of gp120^{HIV-2}.

Supplementary Fig. 9 panel d:

d Representative local density maps of the V3 loop of gp120^{HIV-2}.

Response Fig. 1

Response Fig. 1 AlphaFold3 structural predictions of CXCR4 assemblies and complexes compared to cryo-EM models.

a Five AF3-predicted models of full-length CXCR4 tetramers, showing limited structural conservation and low predicted interfacial confidence scores (ipTM: 0.22, pTM: 0.36). The N-terminal domain (NTD) loop and C-terminal domain (CTD) helix are highly flexible across models.

b Structural alignment of the CXCR4 tetramer from the AF3-predicted model 3 (gray) with the cryo-EM-derived tetramer (magenta) reveals a striking difference in tetramer assembly, while the monomer aligns well with high accuracy (RMSD = 0.769 Å).

c Five AF3-predicted models of the CXCR4-gp120^{HIV-2} monomeric complex (ipTM: 0.15, pTM: 0.49), displaying poor model conservation and high flexibility in the NTD loop, CTD helix, and the V3 loop of gp120 (highlighted in red).

d Overlay of five CXCR4-gp120^{HIV-2} models, showing gp120 conformational variability relative to the CXCR4 monomer. The V3 loop fails to stably insert into the CXCR4 CRS2 pocket.

e Five AF3-predicted models of the CXCR4-gp120^{HIV-2}-CD4 monomeric complex (ipTM: 0.50, pTM: 0.54), with CD4 shown in green and gp120 V3 loop in red. Similar flexibility is observed in the V3 and CTD regions.

f Overlay of the five CXCR4-gp120^{HIV-2}-CD4 models, showing conserved CD4-gp120 interfaces but conformational variation of gp120 relative to CXCR4.

g Structural alignment of the five CXCR4-gp120^{HIV-2}-CD4 models, highlighting conserved interface between gp120^{HIV-2} and CD4, with RMSD values of 0.475 Å, 0.447 Å, 0.471 Å, and 0.426 Å compared to model 1.

h Superposition of AF3 models of the CXCR4-gp120^{HIV-2}-CD4 complex with the cryo-EM structure shows good agreement in the gp120-CD4 interface (RMSD = 0.819 Å, compared to model 1). The V3 loop containing the GFKF motif, which inserts into the CRS2 pocket in the cryo-EM model, is not accurately predicted by AF3.

2. The HIV-2 gp120: CXCR4 ratio looks very odd by SDS-PAGE in Extended Data Fig. 1.c and d, with too much gp120 and almost no CXCR4, inconsistent with the stoichiometry revealed by the structures. In the meanwhile, Extended Data Fig. 10 showed a reasonable ratio for HIV-1 gp120 and CXCR4, even though the complex is supposed to have lower binding affinity.

Thank you for your questions. We have updated the SDS-PAGE gels for the CXCL12-CXCR4, CXCR4-gp120^{HIV-2}, and CXCR4-gp120^{HIV-2}-CD4 complexes in **Supplementary Fig. 3, panel a**, and **Supplementary Fig. 5, panels a and b**, respectively. We noticed that the CXCR4 bands appeared faint and diffuse, which we suspected was due to glycosylation. To test this, we treated both CXCR4 tetramer and monomer samples with PNGase F and observed that the bands shifted to lower molecular weights and became more distinct (**Supplementary Fig. 1, panel f**). PNGase F is known to efficiently remove almost all N-linked oligosaccharides, confirming that CXCR4 is likely N-linked glycosylated.

To examine the stoichiometry of the gp120^{HIV-2}-CXCR4 complex, we used native PAGE and mass photometry. Mass photometry showed that the molecular weight of the CXCR4-gp120^{HIV-2} complex (approximately 365 kDa) was higher than that of the CXCR4 tetramer alone (272 kDa), indicating the binding of one or two gp120 molecules (each ~58.2 kDa) per tetramer (**Supplementary Fig. 5, panels e-g**). This result is consistent with the stoichiometry observed in our cryo-EM dataset. In addition, band shifts were clearly visible in the native PAGE analysis (**Supplementary Fig. 5, panels c and d**), supporting the formation of these complexes. Overall, these data indicate that the gp120^{HIV-2}: CXCR4 binding ratio is reasonable and consistent with our structural findings. These results have been updated on revised manuscript regarding methods, **Supplementary Fig. 1, panel f**, **Supplementary Fig. 3, panel a**, and **Supplementary Fig. 5**.

Methods:

“Native PAGE

The expression and purification of CXCR4, CXCL12-CXCR4, HIV-2 gp120 and CD4-ECD were same as described above except that eluted CXCR4 was subjected to size-exclusion chromatography (SEC) using a Superdex 200 Increase 10/300 GL column (GE Healthcare). Moreover, CXCR4 (calculated as monomer) was assembled with HIV-2 gp120 or gp120-CD4 at a 1:6 molar ratio to guarantee the maximum assembly efficiency. Purified CXCR4, CXCL12-CXCR4, CXCR4-gp120 and CXCR4-gp120-CD4 proteins (totally 3µg each) were loaded into native PAGE (4 to 16%, Thermo Scientific). The gel was run in low temperature (4°C) with 150 V Constant for 60 mins and then 250 V Constant for another 60 mins. The gel was fixed in the solution (40% methanol, 10% acetic acid) and heated for 45 seconds. After shaking for 15 minutes at room temperature, destain solution (8% acetic acid) was added and the gel was heated for 45 seconds. The bands were recorded by using Gel Doc XR+ imaging system (Bio-Rad) when the desired background is obtained.

Mass photometry

Mass photometry experiments were performed using a Refeyn TwoMP instrument. A pre-assembled 6-well sample cassette (Refeyn) was placed at the centre of a clean sample carrier slide (Refeyn), with each well designated for an individual measurement. To establish the focal point, 15 µL of freshly prepared SEC buffer (25 mM HEPES pH 7.5, 150 mM NaCl, 0.001% LMNG, 0.0001% CHS and 0.00033% GDN) were added to the well. The focus was determined and maintained throughout the measurement using an autofocus system based on total internal reflection. Purified CXCR4, CXCL12-CXCR4, CXCR4-gp120 and CXCR4-gp120-CD4 proteins, prepared in the same SEC buffer, were initially diluted to 200 nM and 3 µL of the diluted sample was added to the buffer drop, achieving a final protein concentration of 33.3 nM. Once the autofocus stabilised, movies were recorded for 60 seconds. Data were collected using Refeyn AcquireMP (version 2024.1.1.0) and analysed with Refeyn DiscoverMP (version 2024.1.0.0). Contrast-to-mass calibration was carried out using a BSA protein standard (Sigma), which contained BSA monomers and dimers with molecular masses of 66.5 and 132 kDa, respectively. Statistical analysis was conducted using DiscoverMP, where Gaussian fitting was applied to distribution peaks to determine the average molecular mass of each component. Plotting was carried out using GraphPad Prism 10.

Deglycosylation assay

PNGase F (P0704S, NEB) was used for removing N-linked oligosaccharides from CXCR4. Briefly, 0.42 µg CXCR4 proteins (tetramer or monomer) were mixed with 1× GlycoBuffer 2 and 0.5 µl PNGase F to a total reaction volume of 20 µL. Reaction without PNGase F served as negative controls. After incubation at 37°C for 1.5 hour, the separation of reaction products was analyzed by western blot experiments using HRP-conjugated antibodies against Flag-tag (1:2,000-fold dilution (v/v)). The bound antibodies were visualized by using an enhanced chemiluminescence reagent and quantified by densitometry using a ChemiDoc MP imaging system (Bio-Rad).”

Supplementary Fig.1 panel f

f Western blot analysis with and without PNGase F treatment confirms glycosylation of CXCR4 in both monomeric and tetrameric forms.

Supplementary Fig. 3 panel a:

a Size-exclusion chromatography (SEC) and SDS-PAGE confirm successful co-expression of the CXCR4-CXCL12 complex at a 1:1.5 molar ratio (monomer basis).

Supplementary Fig. 5:

Supplementary Figure 5. Protein purification and biophysical of CXCR4-gp120^{HIV-2} and CXCR4-gp120^{HIV-2}-CD4 complex.

a Size exclusion chromatography (SEC) and SDS-PAGE results for the CXCR4-gp120^{HIV-2} complex indicate a shift towards the CXCR4 peak.

b SEC and SDS-PAGE of the CXCR4-gp120^{HIV-2}-CD4 complex demonstrate a significant shift towards the peaks corresponding to either CXCR4 or gp120^{HIV-2}-CD4.

c Blue native PAGE (BN-PAGE) analysis reveals the formation of CXCR4-gp120^{HIV-2} complex.

d BN-PAGE analysis of CXCR4 complexes with gp120^{HIV-2}-CD4 reveals the formation of distinct complexes.

e-g Mass photometry analyses of ligand-free or ligand-bound complexes reveal molecular masses corresponding to CXCR4 tetramer (~272 kDa, e), CXCR4-gp120^{HIV-2} (~365 kDa, f), and the ternary CXCR4-gp120^{HIV-2}-CD4 complex (~417 kDa, g), with buffer blank signals highlighted by asterisks.

3. These structures have not provided answers to some important questions in the HIV field, such as, what are structural determinants for co-receptor usage (R5 vs X4, or R5X4), and how the virus manages to switch coreceptor. Again, it may have to do with the quality of the cryo-EM maps that could not provide enough information at high resolution.

Thanks for your valuable insights. After reviewing the modeled CXCR4-gp120^{HIV-1} complex, we gained insights into how specific mutations, such as D320R and S306R, could contribute to co-receptor switching. We believe this structural model provides a valuable framework for understanding the molecular basis of co-receptor usage. In the revised manuscript, we have added a new section to discuss these findings in greater detail, along with a corresponding new **Fig. 5**.

Results:

“Potential mechanism of co-receptor switch

It is well established that the amino acid sequence of V3 determines HIV-1 co-receptor usage-whether the virus binds to CCR5 (“R5 viruses”), predominantly infecting macrophages, or to CXCR4 (“X4 viruses”), primarily infecting T cells²⁹⁻³¹. We revisited the modeled CXCR4-gp120^{HIV-1} complex to investigate how specific mutation could drive the transition from an R5 to an X4 phenotype. Previous studies have reported that mutations such as D320R or S306R in the V3 loop are associated with enhanced CXCR4 binding. In our structural model, this aspartate-to-arginine substitution enable a salt-bridge with E31, located in the N-terminus of CXCR4, which strengthens the gp120-CXCR4 interaction (**Figs. 5a, b**). Similarly, the S306R introduces a positively charged residue that can form a cation- π with F189_{CXCR4}, further stabilizing the interaction (**Fig. 5c**). Residues 323 and 324 in the V3 loop, which are typically negatively charged or uncharged and associated with CCR5 usage, may form hydrogen bonds or salt bridges with D181 in the ECL2 region of CXCR4 when mutated to positively charged residues (**Fig. 5d**). Notably, these residues are not conserved in CCR5, highlighting their roles in co-receptor specificity and V3 loop recognition. Our structural model offers a mechanistic explanation for how these site-specific mutations influence co-receptor preference, consistent with prior experimental observations^{27,28,30}. It also provides fundamental structural evidence for how a single mutation can enable the switch from CCR5 to CXCR4 usage, contributing to our understanding of viral tropism and pathogenesis.”

Figure 5:

Figure 5. Structural basis for enhanced CXCR4 binding by HIV-1 gp120 V3 loop mutants.

a Structural model of HIV-1 gp120 (green) bound to CXCR4 (magenta), overlaid with CCR5-bound gp120^{HIV-1} from PDB: 6MEO (gray), highlighting the V3 loop and positions of key interface residues.

b-d Close-up comparisons of wild-type (left panels) and mutant (right panels) V3 loop residues and their interactions with CXCR4. **b** The D320R mutation introduces a bifurcated salt bridge with CXCR4 residues E31 (N-terminus) and E277 (ECL3), enhancing electrostatic complementarity. **c** The S306R mutation enables a cation- π interaction with CXCR4 F189. **d** Double mutation G323R/D324R allows formation of hydrogen bonds or salt bridges with CXCR4 D181 and D182 in ECL2.

Minor points:

1. Writing of the manuscript needs to be improved. The authors should be mindful of all the terms they use. For example, in the sentence -“The subsequent conformational changes in the fusion peptide gp41 bring the viral and host membranes together, promoting their fusion”. Most likely, they meant “the fusion subunit gp41”. Fusion peptide means a completely different thing in the viral entry field. Also, not sure why the CXCR4 tetramer is called CXCR4 complex? Complex with what?

Thank you for pointing out these errors. In the revised manuscript, we have replaced “CXCR4 complex” (from the CXCR4 tetramer structure) with “CXCR4 structure”.

2. It was first stated that “The N-terminus of CXCR4 acts as the chemokine recognition site 1 (CRS1), engaging with the globular core of CXCL12”, and then “Similar to the CXCR4 tetramer complex alone, the N-terminal residues (1-25) of CXCR4 in the CXCL12-CXCR4 complex are not visible, likely due to inherent flexibility.” One of the statements has to be incorrect.

Thank you for your insightful comment. The statement “The N-terminus of CXCR4 acts as the chemokine recognition site 1 (CRS1), engaging with the globular core of CXCL12” refers to the “two-site” model of chemokine binding, and we have added the appropriate citations to support it. In our CXCR4-CXCL12 complex structure, the N-terminal residues 1 to 25 of CXCR4 are unresolved, which is consistent with all previously reported structures, including monomeric, dimeric, trimeric, and tetrameric forms.

In the monomeric CXCR4-CXCL12-Gi complex structures (PDBs 8U4O and 8K3Z), the resolved portion of the N-terminus begins at M24, which is consistent with our own data. In our structures, the CXCR4 apo form and the CXCR4-CXCL12 complex both begin at K25, while the CXCR4-gp120^{HIV-2} and the CXCR4-gp120^{HIV-2}-CD4 complexes begin at E26. A similar trend is observed in other CXCR4 structures. For example, the CXCR4-vMIP-II complex (PDB 4RWS) starts at S23. Structures bound to small molecule antagonists, such as HF51116, AMD070, and AMD3100 (PDBs 8ZPL, 8ZPM, and 8ZPN), start at A34. Other complexes, such as those with IT1t (PDB 3ODU) and CVX15 (PDB 3OE0), begin at P27 and K25, respectively. The recently reported CXCR4 tetramer structure (PDB 8YU7) also begins at E26.

Together, these data suggest that the N-terminus of CXCR4 is often unresolved in both ligand-bound and apo states. This includes complexes with chemokines (monomeric and tetrameric), viral peptides such as vMIP-II, small molecule antagonists (monomeric and dimeric), and viral glycoproteins such as HIV-2 gp120 in tetrameric form. However, we cannot conclude that the N-terminus of CXCR4 is inherently flexible, as the “two site” model of chemokine binding, supported by NMR structures of the CXCR4 N-terminus in complex with CXCL12 (PDBs 2N55 and 2K04), suggests that it may adopt a defined conformation under certain conditions. We have removed the phrase “likely due to inherent flexibility” and revised the main text to reflect the updated discussion in the revised manuscript.

Results:

“Two distinct assembly mechanisms of CXCL12-CXCR4 complex formation”

The N-terminus of CXCR4 acts as the chemokine recognition site 1 (CRS1), engaging with the globular core of CXCL12, while the seven transmembrane helices of CXCR4 form chemokine recognition site 2 (CRS2), which accommodates the insertion of the N-terminus of CXCL12 into a pocket ³⁸⁻⁴⁰. In our structure, similar to the CXCR4 tetramer alone, the N-terminal residues (1-25) of CXCR4 in the CXCL12-CXCR4 complex are not visible, which is consistent with all reported CXCR4 ligand-bound and apo states, including complexes with chemokines (PDBs: 8U4O and 8K3Z)^{25,34}, viral peptides (e.g. vMIP-II, PDB: 4RWS)⁴¹, small-molecule antagonists (PDBs: 8ZPL, 8ZPM, 8ZPN)⁴².”

3. “Notably, the monomer interfaces in our cryo-EM structures and those in the crystal structures are strikingly different (Fig. 1b). Recent studies have also shown that CXCR4 can exist in monomeric and trimeric forms ^{25,28}, highlighting the highly dynamic nature of CXCR4 assembly. This structural flexibility may be linked to its functional diversity, as different CXCR4 conformations could correspond to distinct cellular outcomes.” Could it be caused by different conditions used for different studies?

We thank the reviewer for this helpful suggestion. Indeed, differences in CXCR4 oligomerization may be influenced by its functional state (e.g., CXCL12-Gi-bound active state vs. inactive state),

the structural determination method (crystallography vs. cryo-EM), and variations in detergent and critical micelle concentration (e.g., DDM or LMNG vs. GDN) used during purification. Additionally, the lipid environment associated with the expression system (e.g., insect cells vs. mammalian cells) may also contribute. We have addressed this point in detail in our response to Reviewer #2, specific comment 1, as they raised a similar question (Pages 14-19).

4. “Our structures offer valuable insights for the development of vaccines and therapeutic agents targeting both HIV-1 and HIV-2 infections. One promising strategy is to design neutralizing antibodies that target the V3 loop of HIV-1 and HIV-2, focusing specifically on the GPGR motif in HIV-1 and GFKF motif in HIV-2”. The authors need to get themselves more familiar with the HIV literature (or at least the manuscript should be looked at by an HIV expert). There are many antibodies against the V3 crown (GPGR/Q) and they are not useful for vaccine development.

Thank you for your valuable advice. Admittedly, we need to be more familiar with the HIV field. We made an effort to read more literature to deepen our understanding, and talk about this manuscript with Peijun Zhang and Serge Bouaziz who are working on HIV field for more advice. In the meantime, we have removed the sentence “One promising strategy is to design neutralizing antibodies that target the V3 loop of HIV-1 and HIV-2, focusing specifically on the GPGR motif in HIV-1 and GFKF motif in HIV-2” to make it more accurate and reliable.

5. “gp120^{HIV-2} (residues 23-501) were fused”. Which HIV-2 strain was used? Is it a full-length gp120?

Thank you for pointing this out. The HIV-2 gp120 used in this study is a biological variant of HIV-2, referred to as HIV-2/vcp. This variant was derived from the HIV-2/NIH isolate and is capable of efficiently utilizing CXCR4 independently of CD4. The sequences of gp120^{HIV-1} and gp120^{HIV-2} used for complex assembly have been included in **Supplementary Fig. 8, panels c and d**, we have also updated the methods, and the references accordingly in the revised manuscript.

Methods:

“Protein expression and purification

For recombinant expression in 293F cells, the gene of full-length human CXCR4 was cloned into pcDNA3.4 vector with a C-terminal Flag tag. Genes of human CXCL12, CD4-extracellular domain (ECD, residues 26-388), gp120^{HIV-1} (from the 92BR020 strain, residue 30-505, Uniprot: B2LT42)²¹ and gp120^{HIV-2} (from the VCP strain, residues 23-501, Uniprot: Q70145)^{50,59} were fused with interleukin-2 (IL-2) secretion signal peptides at their N terminus and His tags at their C terminus, and then cloned into pcDNA3.4 vector separately.”

Supplementary Fig. 8, panels c and d:

c HIV2-gp120 (UniProt ref.:B2LT42)

MYRMQLLSICIALSLALVTNSKQFVTVFYGIPAWRNASIFLCATKNRDTWGTIQCLPDNDYQEIALNVTEAFDAWNNVT
TEQAVEDVWNLFETSIPKPCVKLTPLCVAMNCTRNMTTSTGTTDTQNIITINDTSPCVRADNCTGLKEEEMVDCQFNMTGL
ERDKRKQYTEAWYKDVICDNTSSRSKCYMNH CNTSVITESCDKHYWDAMRFRYCAPPGFALLRCNDTNYSGFAPN
CSKVVAATCTRM METQSSTWFGFNGTRAENRTYIYWHGKNNRTIISLNNFYNLTMHCKRPGNKTVLPIMSGFKFHSPV
INKKPRQAWCWFKGEWKEAMQEVKETLAKHPRYKGNRSRTENIKFKAPGRGSDPEAAYMWTNCRGEFLYCNMTWFL
NWVDNRTGQKQRNYAPCHIRQIINTWHRVGNVYLPREGELTCNSTVTSIIANIDTGDQTDITFSAEVAELYRLELGDYK
LVEITPIGFAPTSVKRYSSAHQRHTRHHHHHH*

IL2-signal peptide – gp120^{HIV-2}(vcp strain, residues 23-501) – 6×His

d

HIV1-gp120 (UniProt ref.:Q70145)

MYRMQLLSICIALSLALVTNSNLWVTVYYGVPVWKEATTLFCASDAKAYKAEVHNWATHACVPTDPNPQEIVLENTVE
NFMNWKNMVEQM HEDIISLWDQSLKPCVKLTPLCVTLNCIDLNNSTNNNNSSGVKTGIDKGEIKNCSFNNTTTSVKDKEK
KEYALFYNL DVVQIGNDNTSYRLTSCNTSVITQACPKVTFEPIPIHYCTPAGYAILKCNCGKFKFNGTGPCTNVSTVQCTHGI
KPVVSTQLLLNGSLAEEDIVIRSENLTNNAKTIIVQLKDPVDINCTRPNNNTRKSIHIGPGRAFAYATGDIIGDIRQAHCNLSRA
QWNDTL SKIVTKLREQFENKTIKFQPPSGGDEIVFHSFNCGGEFFYCNTTQLFNSTWNTNTEGTSNTTGNDTITLPCRI
KQIVNMWQEVGKAMYAPPIKGIKICSSNITGLLLTRDGGNNEMNTTEIFRPGGGMDMRDNWRSELYKYKVVRIEPLGIAPT
RAKRRVVQREKRHHHHHH*

IL2-signal peptide – gp120^{HIV-1}(isolate 92BR020, residues 30-505) – 6×His

c The HIV-2 gp120 construct from the VCP strain, comprising residues 23–501, was used in this study (UniProt: B2LT42).

d The HIV-1 gp120 construct from the 92BR020 strain, comprising residues 30–505, was used in this study (UniProt: Q70145).

References:

50. Endres, M. J. *et al.* CD4-independent infection by HIV-2 is mediated by Fusin/CXCR4. *Cell* **87**, 745-756 (1996).

59. Lin, G., Lee, B., Haggarty, B. S., Doms, R. W. & Hoxie, J. A. CD4-independent use of Rhesus CCR5 by human immunodeficiency virus Type 2 implicates an electrostatic interaction between the CCR5 N terminus and the gp120 C4 domain. *J Virol* **75**, 10766-10778 (2001).

6. Representative density for gp120 should be shown in Extended Data Fig. 5. Density for gp120 is very bad in Extended Data Fig. 8.

Thanks for your suggestion. In **Supplementary Fig. 6, panel d**, we present the cryo-EM density at the interface between CXCR4 CRS2 and the V3 loop of gp120^{HIV-2}, showing clearly defined side chains in both regions (related to Fig. 3 panel d). This is accompanied by an overall density map of gp120 in **Supplementary Fig. 6, panel e**. Additionally, **Supplementary Fig. 9, panel d** displays local density maps of the V3 loop in the CXCR4-gp120^{HIV-2}-CD4 complex, illustrating a good fit of side chains at the CXCR4-gp120^{HIV-2} interface.

Supplementary Fig. 6, panels d and e:

d Representative local density maps of CXCR4-gp120HIV-2 interface.
e Overview density map of gp120^{HIV-2}.

Supplementary Fig. 9, panel d:

d Representative local density maps of the V3 loop of gp120^{HIV-2}.

NCOMMS-25-11063A Response to Reviewers

Reviewer #1 (Remarks to the Author):

The authors have addressed all my concerns.

Thank you for your valuable review comments which helped us to improve our manuscript.

Reviewer #2 (Remarks to the Author):

The authors made a great effort to address all raised comments, resulting in a much-improved manuscript. While the lack of functional validation with mutagenized CXCR4 remains a concern, the structural information contained in the manuscript provides important advancement to the research field. There is no additional comments from this reviewer.

Thank you for your positive feedback and support, we hope our structural findings provide a strong basis for future functional studies of CXCR4.

Reviewer #3 (Remarks to the Author):

The authors have addressed my previous comments.

Thank you for your valuable review comments which helped us to improve our manuscript.